# Role of the nuclear membrane protein Emerin in front-rear polarity of the nucleus

Paulina Nastały 1, Divya Purushothaman[1], Stefano Marchesi[1], Alessandro Poli[1], Tobias Lendenmann[2], Gururaj Rao Kidiyoor[1], Galina V. Beznoussenko[1], Stefania Lavore[1], Orso Maria Romano[1], Dimos Poulikakos 2, Marco Cosentino Lagomarsino[1,3], Alexander A. Mironov[1], Aldo Ferrari[2,4] & Paolo Maiuri 1✉

Cell polarity refers to the intrinsic asymmetry of cells, including the orientation of the cytoskeleton. It affects cell shape and structure as well as the distribution of proteins and organelles. In migratory cells, front-rear polarity is essential and dictates movement direction. While the link between the cytoskeleton and nucleus is well-studied, we aim to investigate if front-rear polarity can be transmitted to the nucleus. We show that the knock-down of emerin, an integral protein of the nuclear envelope, abolishes preferential localization of several nuclear proteins. We propose that the frontally biased localization of the endoplasmic reticulum, through which emerin reaches the nuclear envelope, is sufficient to generate its observed bias. In primary emerin-deficient myoblasts, its expression partially rescues the polarity of the nucleus. Our results demonstrate that front-rear cell polarity is transmitted to the nucleus and that emerin is an important determinant of nuclear polarity.

[1] FIRC (Fondazione Italiana per la Ricerca sul Cancro) Institute of Molecular Oncology (IFOM), Milan 20139, Italy. [2] Laboratory of Thermodynamics in Emerging Technologies, Department of Mechanical and Process Engineering, ETH Zurich, Sonneggstrasse 3, CH-8092 Zurich, Switzerland. [3] Department of Physics, University of Milan, via Celoria, 16 20133 Milan, Italy. [4] EMPA, Swiss Federal Laboratories for Materials Science and Technology, 8600 Dübendorf, Switzerland. ✉email: paolo.maiuri@ifom.eu

Cell polarity is defined as an intrinsic asymmetry observed in the structural orientation of the cytoskeleton, mainly due to actin filaments and microtubules[1]. It is manifested in cell shape and structure as well as distribution of proteins and cellular organelles. Cellular polarity is crucial in many biological processes, such as morphogenesis, differentiation, proliferation, and migration[2]. Particularly, migration is a fundamentally polarized process that requires the organization of the cell machinery along a front-to-rear axis. Various migratory cell types display a characteristic morphology with a protruding front, at the opposite of a retracting trailing edge[3]. This so-called, front-rear polarity is essential and dictates the direction of movement[4,5]. It also parallelizes the polarization of intracellular compartments as well as signaling cascades[3]. However, it is still unknown if and how front-rear polarity of a cell is transmitted to the largest cellular organelle, the nucleus. Seminal studies reported a tension-induced basal-to-apical polarization of lamin A/C in mouse embryonic fibroblasts[6,7]. However, it has not been further investigated how much this impacts other nuclear envelope (NE) proteins or the nuclear interior.

As it has been previously shown that the nucleus exhibits radial organization[8,9], in this study, we aim to analyze it along the front-to-rear axis. To that end, we systematically and quantitatively study spatial distribution of various components in front-to-rear polarized cells. Our findings reveal that the asymmetric organization of the cell can be transmitted to the nucleus.

## Results

**Distribution map preparation.** To evaluate a possible transmission of polarity from cytoskeleton to nucleus we needed a cell population with a clear front-rear polarity in which to test this hypothesis. We therefore plated human hTERT RPE-1 (RPE1) cells on fibronectin-coated micro-patterned lines 10 μm in width[10]. Cells spreading on such substrates acquire an elongated shape, develop a spontaneous front-rear polarity and randomly migrate in 1D[11] (Fig. 1a). This condition is considered to partially recapitulate the in vivo environment by mimicking the directional orientation of extracellular matrix fibres that occur within

tissue[12]. Afterward, we developed a method to quantitatively analyze spatial protein distribution by combining images of multiple cells spread on the lines[13]. Essentially, images were oriented by the Golgi apparatus, here used as a marker of directionality[4], filtered, registered for the positioning of the nucleus[14], and analyzed (Fig. 1b–g). As expected, the Golgi apparatus orientation determined the corresponding positioning of the microtubule-organizing center (MTOC), both towards the putative direction of motion (Fig. 2a, b). This bias was clearly lost in a randomly oriented map (Supplementary Fig. 1a, b) proving the validity of our method. To understand whether polarity could be transmitted from the cytoskeleton to the nucleus, we started to systematically explore the preferential distribution of proteins at the interface between these two cellular compartments: the nuclear envelope (NE).

**Emerin as a candidate involved in polarity transmission.** Emerin (EMD), an integral membrane protein of the of the NE[15], has been reported to be present at both, the inner (INM) and the outer nuclear membrane (ONM)[16], hence potentially enabling polarity transmission from outside the nucleus to the inside. It was shown to play diverse roles, including chromatin tethering, cellular polarity organization, cell signaling, gene expression, and mechanotransduction[15–21]. Interestingly, as first evidence of nuclear polarity, we observed a significant frontal enrichment of EMD in front-rear polarized cells (Fig. 2c; Supplementary Fig. 2a, b) Because its peculiar localization at both sides of NE could be determinant for nuclear polarity transmission, we decided to systematically compare protein map distributions in control and EMD knock-down cells.

**Cytoplasmic effects of EMD knock-down.** We first observed that the effects of EMD knock-down (Supplementary Fig. 2i) were not restricted to the nucleus. Indeed, it induced a general rearrangement of the cytoskeleton: the nucleus was positioned peripherally within the cell, and the cells had much shorter retractable tails than the control cells (Fig. 2a, b). Both F-actin and focal adhesion distributions were strongly perturbed (Supplementary

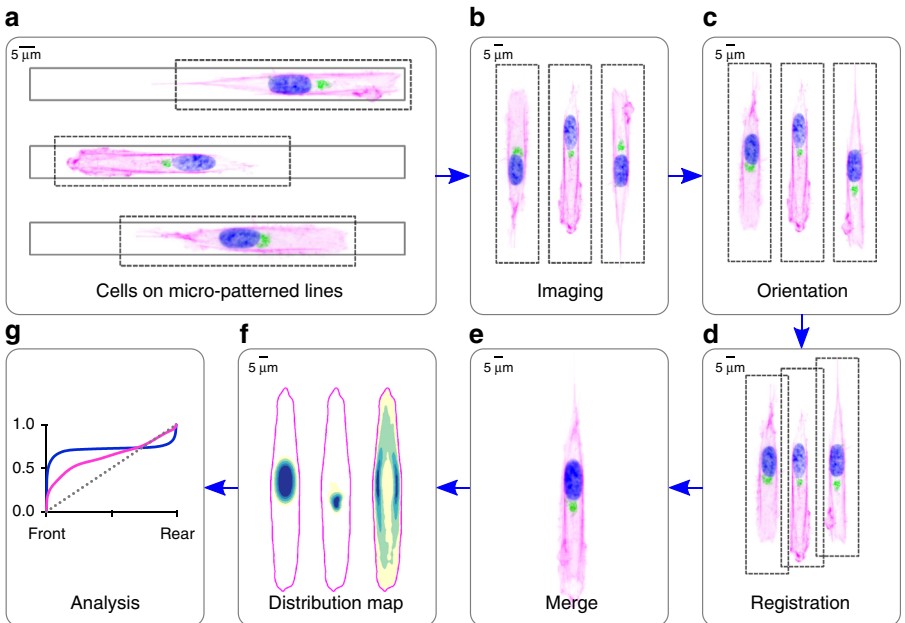

**Fig. 1 Distribution map preparation scheme. a** Representative cells migrating on micro-patterned lines. **b** Random images of single cells. **c** Orientation of single cells using Golgi signal as a marker of directionality **d** Nuclei registration. **e** Merged image of multiple cells from the dataset. **f** Color-coded distribution maps of the nucleus, Golgi and actin. **g** Normalized density plot of the front-to-rear distribution.

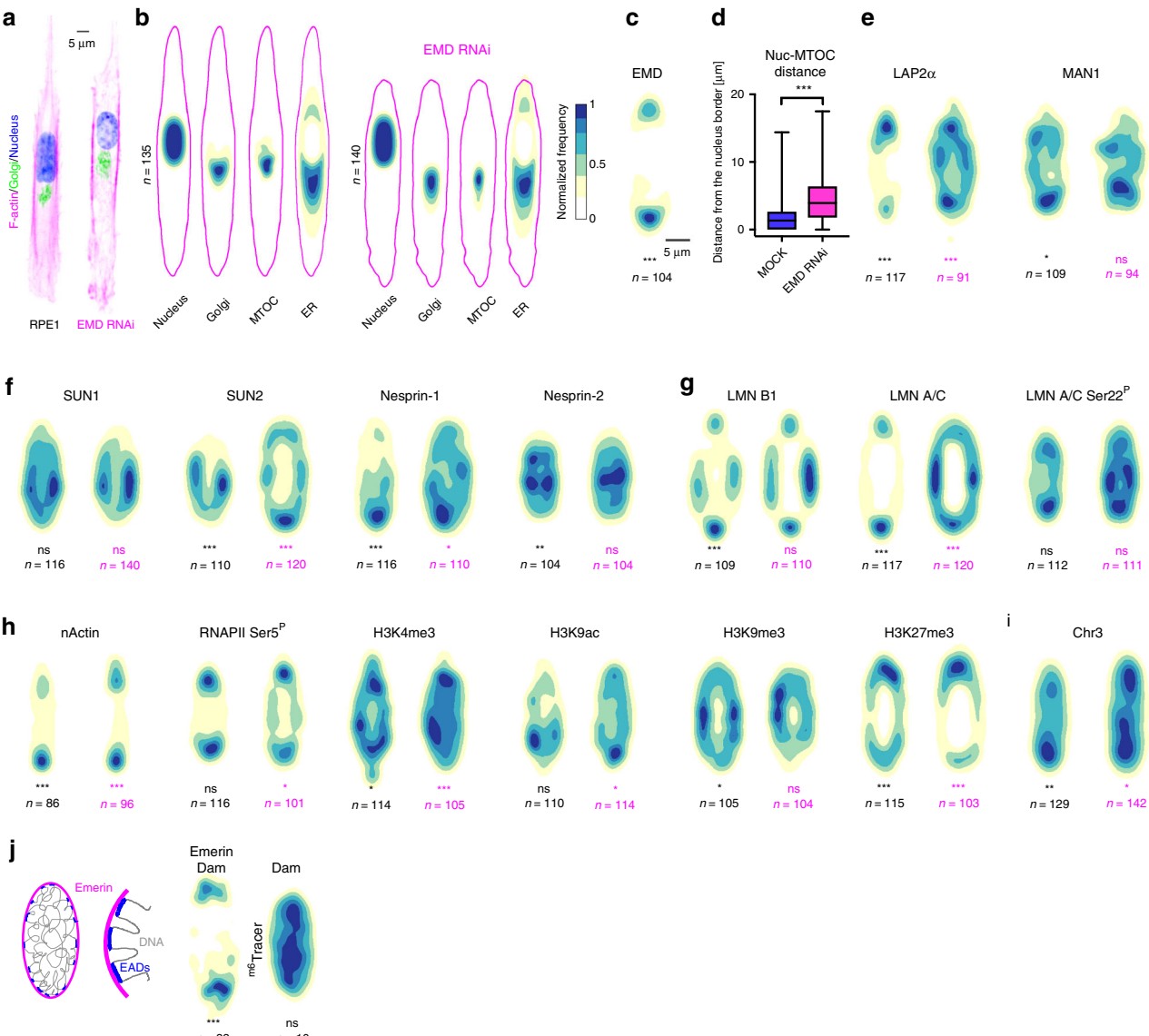

**Fig. 2 Mapping the front-rear polarity in the nucleus.** To facilitate the comparison, the distribution maps of control and EMD knock-down cells are presented next to each other. **a** Single control and EMD knock-down RPE-1 cells on 10 μm micro-patterned lines. **b** Distribution maps of the nucleus, Golgi, MTOC, and ER in control (left panel) and EMD knock-down (right panel) cells. The magenta outline represents the F-actin border of the averaged cells. Normalized frequency color-coding used to prepare all the maps in the study. **c** Distribution map of EMD, $P = 6 \times 10^{-13}$, two-sided Kolmogorov–Smirnov. **d** Distance between the border of the nucleus and MTOC, $n_{MOCK} = 135$, $n_{EMDRNAi} = 140$ cells from three independent experiments. The boxes represent the mean values and the line in the box represents median. Whiskers represent the minimum and maximum values. $P = 2.3 \times 10^{-12}$, two-sided Kolmogorov-Smirnov test (**e**) Distribution maps of LEM domain-containing proteins in control (left, $P_{Lap2\alpha} = 9.1 \times 10^{-11}$, $P_{MAN1} = 0.010$ two-sided Kolmogorov–Smirnov) and EMD knock-down (right, $P_{Lap2\alpha} = 0.0004$) two-sided Cramer–von Mises test) cells. **f** Distribution maps of LINC-complex proteins in control (left, $P_{SUN2} = 8.6 \times 10^{-6}$, $P_{Nesprin-1} = 1.9 \times 10^{-9}$, $P_{Nesprin2} = 0.005$ two-sided Kolmogorov–Smirnov test) and EMD knock-down (right, $P_{SUN2} = 2.6 \times 10^{-6}$, $P_{Nesprin-1} = 0.026$, two-sided Cramer–von Mises test) cells. **g** Distribution maps of lamins (LMN), LMN A/C Ser22–Ser22 phosphorylated lamin A/C, in control (left, $P_{LMNB1} = 1.4 \times 10^{-11}$, $P_{LMNAC} = 4.4 \times 10^{-16}$, two-sided Kolmogorov–Smirnov test) and EMD knock-down (right, $P_{LMNAC} = 1.4 \times 10^{-10}$ two-sided Cramer–von Mises test) cells. **h** Distribution maps of transcription-related markers, nActin—nuclear actin, RNAPII Ser5–Ser5 phosphorylated RNA polymerase II, in control (left, $P_{nActin} = 3.8 \times 10^{-8}$, $P_{H3K4me3} = 0.011$, $P_{H3K9me3} = 0.030$, $P_{H3K27me3} = 0.00025$, two-sided Kolmogorov–Smirnov test) and EMD knock-down (right, $P_{nActin} = 3.9 \times 10^{-6}$, $P_{RNApolSer5} = 0.042$, $P_{H3K4me3} = 2.9 \times 10^{-11}$, $P_{H3k9ac} = 0.015$, $P_{H3K27me3} = 0.00023$ two-sided Cramer–von Mises test) cells. **i** Distribution map of chromosome 3 in control (left, $P = 0.007$ two-sided Kolmogorov–Smirnov test) and EMD knock-down (right, $P = 0.013$ two-sided Cramer-von Mises test) cells. **j** Graphical representation of the DamID technique (left) that serves to mark emerin-associated domains (EADs). Distribution maps of Emerin-Dam and Dam protein alone, $P = 7.5 \times 10^{-5}$, two-sided Kolmogorov–Smirnov test. ***$P < 0.001$, **$P < 0.01$, *$P < 0.05$, ns—not significant. For each distribution map an exact number of cells from three independent experiments is stated in the figure. Source data are provided as Source Data file.

Fig. 2j). Moreover, the distribution of the Golgi apparatus and MTOC appeared more spread along the cell body, suggesting a systemic relaxation of intra-cellular connections (Fig. 2b). In addition, we noted, as previously reported[22], a ~2.5-fold increase in nucleus-MTOC distance (Fig. 1d).

**Distribution map of LEM domain-containing proteins**. EMD is one of the LEM domain-containing proteins that are involved in chromatin tethering and gene expression regulation[15,23]. Interestingly, other members of the family showed different behaviors. MAN1 (also known as LEM domain-containing protein 3), like EMD, has a transmembrane domain (TM), and is localized at NE. As EMD, in polarized cells, it was enriched towards the cell front, if with a milder bias (Fig. 2e; Supplementary Fig. 2a, c). LAP2α (Lamina-associated polypeptide 2α), in contrast, is lacking TM and is localized in the nucleoplasm. In polarized cells it was enriched towards the cell rear (Fig. 2e; Supplementary Fig. 2a, c). The EMD knock-down had diverse effects as it has changed the distribution of LAP2α, without affecting that of MAN1 (Fig. 2e; Supplementary Fig. 2a, c). It is reasonable to speculate that EMD and MAN1, which behave in a similar fashion, reach the NE through the same route and each of them could be independently biased. LAP2α, on the other hand, competes with EMD for common interactors and its preferential localization is then affected by EMD loss.

**Distribution map of nuclear envelope proteins**. The NE is connected to the cytoskeleton by the LINC complex, which is composed of inner nuclear membrane SUN (Sad1/UNC-84) domain-containing proteins that bind KASH (Klarsicht-Anc1-syne1 homology) domain-containing nesprins at the outer nuclear membrane. SUN proteins interact with lamins at the NE, while nesprins interact with actin, microtubules and intermediate filaments in the cytoplasm, allowing force transmission from the extracellular matrix to the nucleus[23–26]. We observed that in front-rear polarized cells, SUN1 and SUN2 were differentially localized: SUN1 was uniformly distributed, while SUN2 was enriched at the center of the nucleus (Fig. 2f; Supplementary Fig. 2a, d). This finding is consistent with previous observations suggesting that SUN1 and SUN2, rather than being redundant, have different functions[27]. Similarly, while nesprin-1 was enriched at the front of the cell, nesprin-2 was predominantly centrally distributed (Fig. 2f; Supplementary Fig. 2a, d). Interestingly, nesprin-2 has been recently shown to be accumulated in the front of the cell during confined migration in mouse embryonic fibroblasts[28]. Components of the LINC complex were differentially affected by EMD knock-down. SUN1 and nesprin-2 were unchanged, but surprisingly, SUN2 was enriched towards the front of the cell, while nesprin-1, a well-known EMD interactor[15,29], partially lost its bias (Fig. 2f; Supplementary Fig. 2a, d). We then moved inward and considered the nuclear lamina, which is a meshwork of intermediate filaments composed of A- and B-type lamins that provides structural architecture to the nucleus[30]. Lamin B1 is essential, while lamin A has a crucial role in mechanotransduction[31]. Recent seminal studies reported a tension-induced basal-to-apical polarization of lamin A/C in mouse embryonic fibroblasts[6,7]. In front-rear polarized RPE1 cells, lamin A/C and lamin B1 showed specific patterns. Both were non-uniformly distributed, with lamin A/C clearly enriched at the front tip of the nucleus and lamin B1 localized more to its sides. In interphase, a small amount of lamin A/C is phosphorylated. This phosphorylated lamin A/C no longer localizes at the NE but is located at the nuclear interior[32]. The phosphorylated form of lamin A/C, probably loosing interaction with its partners at NE, was uniformly distributed in the nucleus

(Fig. 2g; Supplementary Fig. 2a, e). Interestingly, lamin A/C, but neither lamin B1 nor phosphorylated lamin A/C, was affected by EMD reduction in RPE1 cells (Fig. 2g; Supplementary Fig. 2a, e).

**Distribution map of nucleoplasmic proteins**. EMD was shown to interact with both nuclear and cytoplasmic actin[33], and it is also known for its actin-capping properties[34]. Therefore, we decided to investigate the distribution of nuclear actin (nActin). Indeed, nActin was more frequently distributed at the front tip of the nucleus (similar to EMD), and upon EMD knock-down, its distribution shifted to both tips (Fig. 2h; Supplementary Fig. 2a, f). As nActin was mainly linked to transcription, interacting with all types of RNA polymerases[35], we investigated the distribution of active RNA polymerase II (Ser5$^P$) (Fig. 2h; Supplementary Fig. 2a, f). In control cells, it was equally present at both tips of the nucleus, whereas in EMD-deficient cells, it was more frequently found at the rear tip (Fig. 2h; Supplementary Fig. 2a, f). It was slightly different from active chromatin markers. Whereas H3K4me3 was enriched towards the front in control cells and changed its distribution to the rear upon EMD knock-down (Fig. 2h; Supplementary Fig. 2a, f), H3K9ac was uniformly distributed in the control cells and was shifted to the front in EMD-deficient cells (Fig. 2h; Supplementary Fig. 2a, f). EMD was also shown to be involved in chromatin remodeling by the replacement of H3K9me2,3 with H3K27me3 upon mechanical stress[20]. Therefore, we investigated markers of constitutive (H3K9me3) and facultative heterochromatin (H3K27me3) (Fig. 2h; Supplementary Fig. 2a, f). H3K9me3-showed slight frontal enrichment but occurred predominantly at the sides of the nucleus; its positioning was not affected by EMD knock-down. Instead, H3K27me3 was mainly distributed at the rear, which was even more pronounced in EMD-deficient cells (Fig. 2h; Supplementary Fig. 2a, f). Overlapping distributions of RNAPII (marker of active transcription) and H3K27me3 (an epigenetic mark of facultative heterochromatin) might seem counterintuitive, however, our data does not imply co-localization of these proteins. The rear (or the front) of the nucleus should be considered as a huge "macro-domain", where both regions of heterochromatin and euchromatin could be enclosed. The spatial resolution of our maps, indeed, which should be considered similar to probability distributions, is not sufficient to infer or exclude the co-localization of different chromatin domains. Moreover, the understanding of the specific molecular mechanisms driving the preferential localization, and re-localization upon EMD knock-down, of the different nucleoplasmic components, clearly warrant follow up study.

**Distribution map of chromosome territories**. Afterward, to understand whether cell polarity could also affect genome spatial organization, we studied chromosome localization. Indeed, chromosomes adopt a conserved and non-random arrangement in sub-nuclear domains called chromosome territories (CTs)[36], and moreover, it has been recently shown that their positioning is partially modulated by EMD[21]. In our analysis, chromosomes 12, 18, and 22 were non-uniformly distributed (Supplementary Fig. 3a–c). Chromosome 3, as the only CT, lost its frontal enrichment upon EMD knock-down (Fig. 2i; Supplementary Fig. 2a, g).

**Emerin-associated genomic domains follow the distribution of EMD**. To further elucidate the functional implications of nuclear polarity, we employed an EMD-DamID system. This technology allows the specific tagging of genomic regions that are in molecular contact with EMD[37,38]. We demonstrated that emerin-associated genomic domains (EADs) were more frequently found

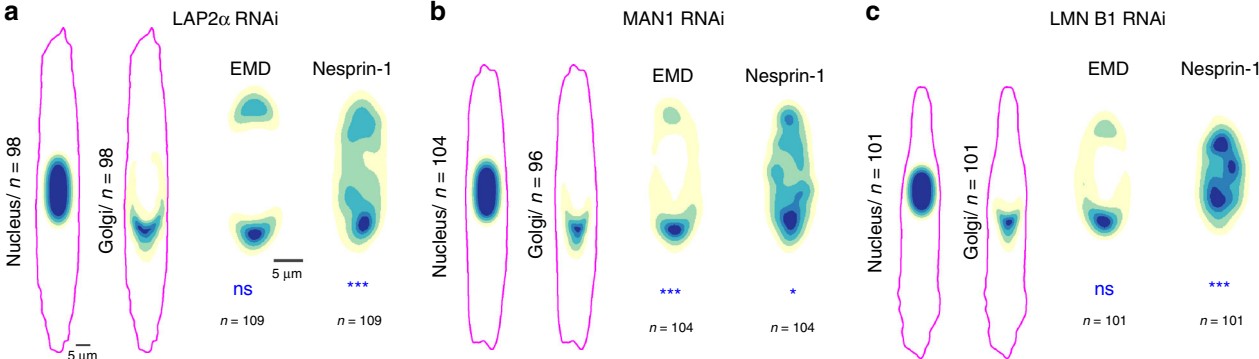

**Fig. 3 Effect of LAP2α, MAN1, and LMN B1 knock-down on nuclear polarity. a** Distribution maps of the nucleus, Golgi, EMD and nesprin-1 in cells with LAP2α knock-down. $P_{Nesprin-1} = 0.0007$, wo-sided Cramer-von Mises test. The magenta outline represents the F-actin border of the averaged cells. **b** Distribution maps of the nucleus, Golgi, EMD and nesprin-1 in cells with MAN1 knock-down. $P_{EMD} = 6.6 \times 10^{-5}$, $P_{Nesprin-1} = 0.011$, two-sided Cramer–von Mises test. The magenta outline represents the F-actin border of the averaged cells. **c** Distribution maps of the nucleus, Golgi, EMD and nesprin-1 in cells with LMN B1 knock-down. $P_{Nesprin-1} = 1.96 \times 10^{-7}$, wo-sided Cramer–von Mises test. The magenta outline represents the F-actin border of the averaged cells. ***$P < 0.001$, **$P < 0.01$, *$P < 0.05$, ns—not significant. For each distribution map an exact number of cells from three independent experiments is stated in the figure. Source data are provided as Source Data file.

at the front side of the nucleus, confirming that the observed protein bias is also transmitted to the chromatin interacting with it (Fig. 2j; Supplementary Fig. 2a, h).

**Distribution maps in non-patterned cells**. To test that our results were not induced by forcing cells to move on one-dimensional lines we investigated protein distribution in cells plated on fibronectin-coated glass (Supplementary Fig. 4a). As polarity is an intrinsic property of cells and cells plated on 2D surface would still spontaneously establish a polarity axis and move in one direction, we applied our method also in this condition. While the cellular and nuclear shape differed from the micro-patterned ones (Supplementary Fig. 4a, b), however after orienting the cells, the maps for preferential protein positions in non-patterned cells were similar to the micro-patterned ones (Supplementary Fig. 4c–g).

**Effect of LAP2α, MAN1, and LMN B1 knock-down also affect nuclear polarity**. Because we found that EMD is necessary for the correct polarity of some components of the nuclear envelope we tested if reciprocally the absence of other proteins could affect its distribution. When we knocked-down other nuclear envelope proteins including LAP2α, and LMN B1, we observed no effect on EMD preferential distribution (Fig. 3a, c; Supplementary Fig. 5a, b). In cells with MAN1-deficiency, EMD frontal enrichment was even more pronounced (Fig. 3b; Supplementary Fig. 5a, b). On the other hand, nesprin-1 distribution was altered in both MAN1 and LAP2α knock-down, however, its frontal localization remained (Fig. 3a, b; Supplementary Fig. 5b). In LMN B1-knock-down condition, nesprin-1 completely lost its frontal enrichment (Fig. 3c; Supplementary Fig. 5b). These results confirm the central role of EMD in nuclear polarity transmission, indeed, EMD is required for the correct localization of other LEM domain-containing proteins, as LAP2α and MAN1, but is not affected but their knock-down. Still, the reduction of both of them impacts on nesprin-1 distribution, as well as the LMN B1 knock-down. These data suggest, first, a possible hierarchy of LEM domain-containing proteins in nuclear polarization and, then, that an integral nuclear envelope is necessary for the establishment of a complete and correct nuclear polarity.

**Migrating cells show EMD enrichment**. We also analyzed RPE1 cells stably transfected with EMD-EGFP, migrating on micro-patterned lines. Importantly, the EMD bias we observed by combining images from multiple fixed cells could also be observed in living cells (Fig. 4a; Supplementary Movie 1).

**EMD plays a role in migration and force transmission**. EMD knock-down significantly perturbed cell migratory properties: cells lacking EMD exhibited increased velocity and persistence (Supplementary Fig. 6a) and impaired chemotaxis efficiency (Supplementary Fig. 6b). We also noted that cells with EMD deficiency had smaller focal adhesions (Supplementary Fig. 6c), and accordingly, they transmitted less force to their substrate[39] (Fig. 4b; Supplementary Fig. 6d), which could explain the observed increase in speed[40,41]. Although our results differ from a previous work examining nesprin-1 and lamin A-deficient cells[42], they indicate that EMD has a crucial role in force transmission and raise further questions about the cytoplasmic role of EMD and the mechanism that could generate its frontal bias at the NE.

**EMD and nesprin-1 localize in the cytoplasm**. Immuno-fluorescent staining for EMD showed that some fraction of the protein localized in the cytoplasm, as was also evident in cells transfected with EMD-EGFP (Fig. 4a). We, therefore, performed transmission electron microscopy (TEM) with immuno-gold labeling to determine the exact localization of EMD. It revealed that EMD is not only present at the INM and ONM, as previously reported, but is additionally found in the cytoplasm on the ER membranes (Fig. 4c). This result is coherent with previous findings reporting initial integration of EMD in the ER membrane, whence it then moves to the ONM, thanks to the continuity between these two compartments, and finally reaches the INM[15,43]. Additionally, the EMD interactor nesprin-1, an actin-binding protein, was present not only at the ONM and in the nucleoplasm but also in the cytoplasm, where it co-localized with actin filaments (Fig. 4c).

**EMD and nesprin-1 bind together in the cytoplasm**. To test whether EMD and nesprin-1 can also interact in the cytoplasm, probably linking the ER to the cytoskeleton, we performed a proximity ligation assay (PLA) between these two proteins (Fig. 4d). As a control, we performed a PLA between lamin B1 and EMD. While the PLA foci between lamin B1 and EMD were mostly restricted to the NE, the ones between nesprin-1 and EMD were observed in the NE and extending into the leading

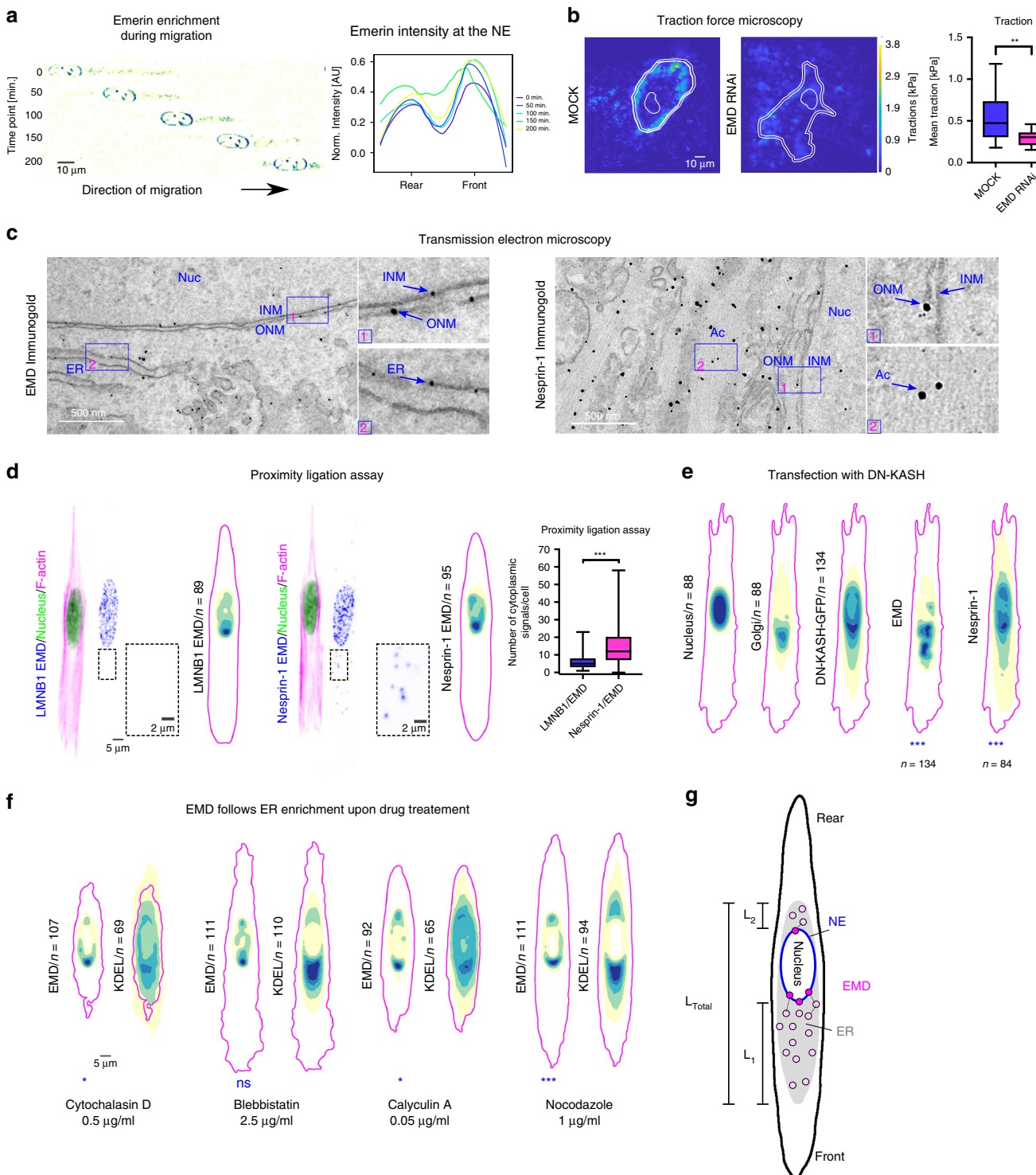

cytoplasmic region (Fig. 4d). Moreover, the preferential distribution maps of protein interactions corresponded to the separate maps of each protein (Fig. 4d).

**Expression of dominant negative KASH domain displaces EMD from NE.** By expressing dominant negative KASH-EGFP (DN-KASH-EGFP) domains that compete with nesprins binding to the SUN proteins, thereby disrupting the LINC complex[24], we found that in addition to nesprin-1, EMD was also displaced from the nuclear envelope. It was indeed present mainly in the frontal cytoplasm of the cell similarly to DN-KASH-EGFP and nesprin-1

(Fig. 4e; Supplementary Fig. 6g). We showed that perturbing the connection between the cytoskeleton and the nucleus, alters nuclear polarity.

**Nuclear polarity persists upon drug treatment.** In order to test whether nuclear polarity can be altered by affecting cytoskeleton components, we incubated micro-patterned cells with different agents including cytochalasin D (prevents actin polymerization[44]), blebbistatin (inhibits myosin[45]), calyculin A (enhances myosin[46]), nocodazole (prevents microtubule polymerization[47]). Although all drug treatments clearly affected both cytoskeleton

**Fig. 4 Functional and structural aspects of nuclear polarity. a** Live cell imaging of frontal emerin enrichment at different time points in migrating cells. **b** Representative images of cells for traction force microscopy and traction quantification ($n_{MOCK}$ = 26, $n_{EMDRNAi}$ = 36 cells from three independent experiments). $P$ value = 0.001,The boxes represent the mean values and the line in the box represents median. Whiskers represent the minimum and maximum values. Kolmogorov–Smirnov test. **c** TEM of immunogold-labeled emerin (left panel) and nesprin-1 (right panel). ONM—outer nuclear membrane, INM— inner nuclear membrane, ER—endoplasmic reticulum, Nuc—nucleus, Ac—actin filaments. **d** Detection of LMNB1/EMD (left panel) and nesprin-1/EMD (right panel) interaction sites, representative images and distribution maps, quantification of cytoplasmic interaction sites ($n_{LMNB1/EMD}$ = 89, $n_{nesprin-1/EMD}$ = 94 cells from three independent experiments). $P$ value = $5.9 \times 10^{-8}$. The boxes represent the mean values and the line in the box represents median. Whiskers represent the minimum and maximum values. Two-sided Kolmogorov–Smirnov test. **e** Distribution maps of the nucleus, Golgi, DN-KASH-GFP, EMD, and nesprin-1 in cells transfected with dominant negative KASH domain. $P_{EMD}$ = $2.3 \times 10^{-11}$, $P_{Nesprin-1}$ = $7.3 \times 10^{-5}$ the two-sided Cramer–von Mises test, \*\*\*$P < 0.001$. **f** Distribution maps of EMD and nesprin-1 in cells treated with cytochalasin D, blebbistatin, calyculin A and nocodazole. $P_{CytochalasinD}$ = 0.035, $P_{CalyculinA}$ = 0.013, $P_{Nocodazole}$ = 0.0004, the two-sided Cramer–von Mises test. **g** Schematic model of EMD enrichment at the NE as a function of ER asymmetry in the cytoplasm, $L_1$— ER frontal length, $L_2$— ER rear length, $L_{Total}$— total ER length. \*\*\*$P < 0.001$, \*\*$P < 0.01$ \*$P < 0.05$, ns—not significant. For each distribution map an exact number of cells from three independent experiments is stated in the figure. Source data are provided as Source Data file.

and ER, as shown by the altered cell shape and KDEL distribution map, the ER and Golgi frontal localization was always preserved (Fig. 4f; Supplementary Fig. 6e). In agreement with the literature, we performed drug treatments using sub-lethal drug concentrations for a short time. These conditions are sufficient to perturb the cytoskeleton but not to destroy the polarity of the cell, and indeed, we detected alterations of nuclear polarity, not its complete loss. Specifically, we observed, that although none of the drugs abolished the EMD frontal enrichment, cytochalasin D and nocodazole significantly decreased it (Fig. 4f; Supplementary Fig. 6f). On the other hand, calyculin A slightly increased the EMD frontal polarization (Fig. 4f; Supplementary Fig. 6f), suggesting that an enhancement in contractility, probably generally increasing cell polarity, also induce an increase of nuclear polarity.

**A mathematical model supports the hypothesis that front-rear bias in the ER could transmit EMD polarity.** Overall, these results suggest that EMD enrichment at the frontal NE originates from the asymmetric distribution of the ER at the leading edge, from which EMD moves to the ONM and further to the INM (Fig. 4g, Supplementary Fig. 6h). To support this view, we developed a minimal mathematical model (Supplementary Information) in which the total amount of EMD entering the ER is assumed to be proportional to the ER surface, and the EMD diffusion in the NE is assumed to be slow enough to be neglected. This simple model reproduces the observed bias of EMD at the NE as a function of the asymmetric distribution of the ER in the cytoplasm and agrees with stochastic simulations (Fig. 4g, Supplementary Fig. 6h, Supplementary Movie 2).

**EMD-deficient primary myoblasts show similar polarity phenotypes as RPE1 cells.** EMD mutations are pathologically relevant because they cause X-linked Emery-Dreifuss muscular dystrophy (EDMD)[48]. This disease affects mainly mechanically active tissues and is associated with progressive muscle wasting and weakening that leads to cardiac disease[49]. To validate our findings, we analyzed the protein distribution in both normal primary skeletal myoblasts from healthy individuals and EMD-deficient (Supplementary Fig. 7a, e) ones from a EDMD patient. Primary myoblasts acquired front-rear polarity on micro-patterned lines and showed differences in cellular organization between normal and EMD-deficient cells, equivalent to the ones observed in RPE-1 cells. Indeed, in EMD-deficient myoblasts, the nucleus was more peripheral, the nucleus-MTOC distance increased and the focal adhesion size decreased (Fig. 5a–c). EMD and nesprin-1 were both enriched towards the front in primary normal myoblasts (Fig. 5d, e), whereas in cells from EDMD

patients, the preferential positioning of nesprin-1 was lost. When EMD-EGFP was expressed in EMD-deficient cells, it localized as the endogenous protein in normal myoblasts (Supplementary Fig. 7b), and the cells partially reverted to a normal phenotype (Fig. 5a–e; Supplementary Fig. 7c, d, f).

## Discussion

Polarity is an intrinsic property of the cells. Almost all adherent cell types, indeed, independently of external cues, are able to self-define an axis of polarity and migrate. This asymmetry implies an asymmetric organization of the cellular architecture, with a non-uniform distribution along the cytoskeleton of proteins, organelles, and tensile stress[1,3]. While the connection between the cytoskeleton and the nucleus is well-studied[24,50], it is unknown if part of the front-rear cell polarity is somehow transmitted to the nucleus.

This study demonstrates that front-rear cell polarity is transmitted from the cytoskeleton to the nuclear envelope and, thus, defines a nuclear polarity. We show that nuclear polarity not only concerns the preferential distribution of proteins at the NE, but also extends to nucleoplasmic proteins and, to some extent, to chromatin. Moreover, we show that EMD, an integral protein of the nuclear envelope, suggested to be involved in mechanotransduction[20,51], is one of the molecular players involved in nuclear polarity transmission. The knock-down of EMD clearly affects the biased distribution along the polarity axis of some components of the nuclear envelope and also in the nucleoplasm. However, our results suggest that there are other molecular components responsible for front-rear polarity of the nucleus, with a complex hierarchy of interactions. For example, the polarity of some elements of the NE, such as nesprin-1 or MAN1, is not affected by EMD reduction. Additionally, also the knock-down of LAP2α, MAN1 or LMN B1 affects the distribution of other NE components, but without perturbing the biased distribution of EMD. This suggests that probably an integral nuclear envelope is necessary for the correct and complete transmission of nuclear polarity.

Notably, we found that protein distribution maps in RPE1 and primary myoblasts, while being similar, did not perfectly match. Therefore, we can conclude that the extent and feature of nuclear polarity are cell-type specific. It is possible that different LINC complex proteins are not only differentially expressed[52] but also distributed in a different manner among various cell types, having either redundant or essential functions[27].

We hypothesize that nuclear polarity is generated by the asymmetric distribution of the ER and that it relies on the continuity between the ER and the outer nuclear membrane. Our results also suggest a possible role of the ER in mechano-sensing. Indeed, EMD and nesprin-1, interacting at the surface of the ER,

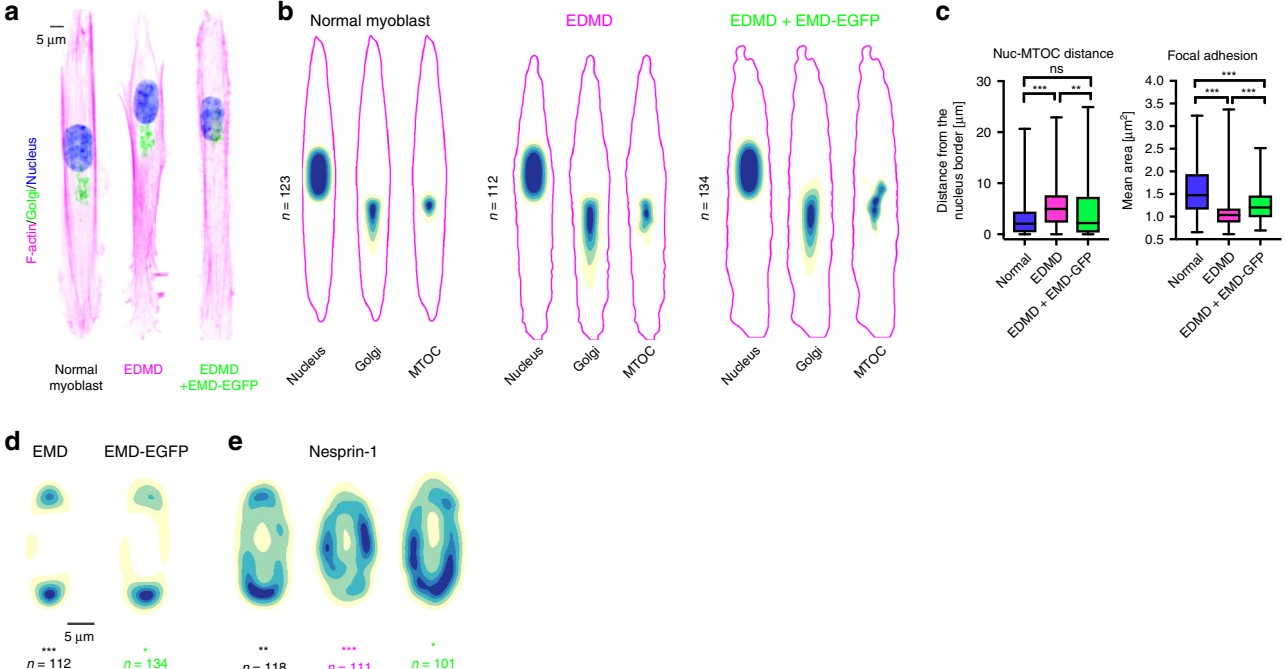

**Fig. 5 Nuclear front-rear polarity rescue in primary myoblasts. a** Single primary normal (left), EDMD (middle) and EMD-EGFP-rescue (right) myoblasts on 10 μm micro-patterned lines. **b** Distribution maps of the nucleus, Golgi, and MTOC in normal (left panel), EDMD (middle panel) and EMD-EGFP-rescue cells (right panel). The outline represents the F-actin border of the averaged cells. **c** Measurement of nucleus-MTOC distance ($n_{Normal} = 123$, $n_{EDMD} = 110$, $n_{EDMD+EMD-EGFP} = 189$ cells from three independent experiments) $P_{normal\ vs.\ EDMD} = 1.3 \times 10^{-5}$, $P_{EDMD\ vs.\ EDMD+EGFP} = 0.0013$. Measurement of size of focal adhesion ($n_{Normal} = 103$, $n_{EDMD} = 87$, $n_{EDMD+EMD-EGFP} = 88$ cells from three independent experiments) $P$ value normal vs. EDMD $= 1.9 \times 10^{-14}$, $P_{normal\ vs.\ EDMD+EMD-EGFP} = 0.00019$, $P_{EDMD\ vs.\ EDMD\ +\ EMD-GFP} = 0.00071$. The boxes represent the mean values and the line in the box represents median. Whiskers represent the minimum and maximum values. Kruskal-Wallis test. **d** Distribution map of EMD in primary normal (left, $P_{EMD} = 3.0 \times 10^{-13}$, two-sided Kolmogorov–Smirnov test) and EMD-EGFP (right, $P_{EMD-EGFP} = 3.4 \times 10^{-5}$, two-sided Cramer–von Mises test). **e** Distribution map of nesprin-1 in primary normal (left, $P = 0.004$, two-sided Kolmogorov–Smirnov test), EDMD (middle, $P = 2.4 \times 10^{-6}$, two-sided Cramer–von Mises test) and EMD-EGFP-rescue (right, $P = 0.04$, two-sided Cramer–von Mises test) myoblasts. *\*\*\*$P < 0.001$, *\*\*$P < 0.01$, *\*$P < 0.05$, ns—not significant. For each distribution map an exact number of cells from 3 independent experiments is stated in the figure. Source data are provided as Source Data file.

could generate a continuous meshwork linking the cytoskeleton, the ER and the nucleus, enabling tension transmission along those various compartments.

The pathological consequence of emerin loss is the Emery Dreifuss muscular dystrophy[53]. While we cannot directly prove that the disease is caused by the loss of nuclear polarity, we show that in primary myoblasts from an EDMD patient nuclear polarity is perturbed. With emerin ectopic expression, we were able to partially rescue the normal phenotype. Mutations in other genes encoding diverse NE proteins including nesprins and lamins lead to the whole spectrum of nuclear envelopathies that are manifested in various tissues[49]. Since we show that an integral nuclear envelope is necessary for the correct and complete transmission of nuclear polarity, we can speculate that it can be also affected in some of these diseases.

Finally, we describe a nuclear polarity as a cellular phenomenon but its role in physiological as well as in pathological conditions have still to be explored.

## Methods

**Cell culture**. Human hTERT-immortalized RPE-1 cell line was cultured in DMEM/HAM's F12 (Bio West, Cat. L0093-500) supplemented with 10% FBS (Euroclone, Cat. ECS0181L). Cells were split every 2–3 days and passaged not more than 6 times. Human primary myoblasts were obtained from Telethon Biobank. Individuals gave written informed consent before undergoing muscle biopsy, in agreement with the Declaration of Helsinki and approved by the ethical committees of the centers, where biological samples were obtained. Normal muscle biopsy (male, catalog#70515) and the one from patient with EDMD (male, catalog# 49031, *Emd* mutation cDNA.539_543delTCTAC) were cultured in DMEM (Lonza, Cat. BE12-614F) supplemented with 20% Fetal Bovine Serum South America (Sigma-

Aldrich, Cat. F9665), 10 μg/ml human recombinant insulin (Sigma-Aldrich, Cat. 11376497001), 25 ng/ml human recombinant fibroblast growth factor (Peprotech, Cat. 100-18B), and 10 ng/ml active human recombinant epithelial growth factor (Vincil-Biochem, Cat. BPS-90201-3). Primary cells were split every 3-4 days and for analysis were taken cells at passage 4–10.

**Micro-patterning**. Micro-patterns of fibronectin-coated lines (10 μm of width) were fabricated using photolithography[13]. The glass surface of the coverslip was activated with plasma cleaner (Harrick Plasma) and then coated with cell repellent PLL-g-PEG (Surface Solutions GmbH, 0.5 mg/mL in 10 mM HEPES). After washing with 1× phosphate-buffered saline (PBS) and deionized water, the surface was illuminated with deep UV light (UVO Cleaner, Jelight) through a chromium photomask (JD-Photodata). Then, coverslips were incubated with an extracellular matrix protein fibronectin (Sigma-Aldrich, Cat. F1056, 25 μg/ml in 100 mM NaHCO₃ pH 8.4). Cells were detached using EDTA 0.02% (Versane, Gibco, Cat. E6758) and left for 16 h to attach on micro-patterned lines.

**Immunofluorescence**. Cells on micro-patterns were fixed with 4%PFA/1× PBS, permeabilized in 0.1%Triton-X/1xPBS, and incubated in blocking solution (1%BSA in 1× PBS). Then, cells were incubated with primary antibodies (as listed in Supplementary Table S1) and proper secondary antibodies (Jackson ImmunoResearch). Nuclei were stained with 4′,6-diamidino-2-phenylindole (DAPI, Sigma-Aldrich Cat. D8417) Cells were mounted with Vectashield® Antifade Mounting Medium (Vector Laboratories, Cat. H-1000-10).

**Chromosome painting**. Fluorescent in situ hybridization was performed using protocol enabling 3D nuclear structure preservation[54]. Briefly, cells were fixed with 4% PFA for 10 min. and immuno-stained with antibody to visualize Golgi apparatus. After post-fixing with 4% PFA for 10 min, the specimens were incubated for at least 60 min. in 20%glycerol/1× PBS, followed by freeze-thawing cycles in liquid nitrogen. The cells were permeabilized in 0.07% Triton-X/1xPBS/0.1 M HCl for 10 min. and DNA was denatured in 50% Formamide/2xSSC (pH = 7.4) for 10 min. Then, chromosome painting probes (Metasystems, Xcyting Chromosome Paints) were added to the specimen, denatured for 3 min. at 75 °C, and

hybridized at least 16 h at 37 °C in hybridization chamber. Afterward, the cells were washed for 10 min. in 2xSSC and 0.1SSC buffers. Nuclei were stained with DAPI (Sigma-Aldrich Cat. D8417) and the samples were mounted in Vectashield® Antifade Mounting Medium (Vector Laboratories, Cat. H-1000-10).

**DamID experiment**. This method was previously used to map genome and nuclear lamina contacts[37,38]. Briefly, 293T cells were transfected with DamID plasmids by calcium phosphate protocol (10 μg DNA for each 10-cm dish). Forty-eight hours later, viral supernatant was collected, 0.45-μm filtered and added to RPE1 cells. For the DamID experiment hTERT RPE-1 cells were co-transfected with either pHIV-TetP-Puro-IRES-EGFP-DPN7 and pl-GW-V5-EcoDam plasmids (control) or pHIV-TetP-Puro-IRES-EGFP-DPN7 and pl-GW-EcoDam-V5-hEMD (visualization of emerin-associated domains)[37]. The plasmids were a kind gift of prof. Bas van Steensel (Division of Gene Regulation, Netherlands Cancer Institute, Amsterdam, Netherlands). After 48 h incubation with lentivirus, medium was changed and the cells were micro-patterned and left for 16 h to attach. The coverslips containing cells were fixed with 4% PFA and immunostained in order to visualize Golgi and/or emerin.

**Transfection and RNAi**. Cells were transfected using Lipofectamine 2000 reagent (Life Technologies, Cat. 11668030). Nuclear actin was visualized by transfecting cells with Actin-Chromobody® plasmid (Actin-nanobody, fused to TagGFP containing a nuclear localization sequence, (ChromoTek). Emerin-pEGFP-C1 plasmid was obtained from Addgene, plasmid #61993[55]. Dominant-negative KASH fused with EGFP plasmid was previously reported[56,57].

RNAi to target EMD, MAN1 and LAP2α was performed using Lipofectamine RNAiMAX reagent (Life Technologies, Cat. 13778030) and 10 nM Human gene specific 27mer siRNA duplexes (Origene) with following sequences:
EMD: 'rCrCrArArGrArArArGrArGrGrArCrGrCrUrUrUrArCrUrCTA'
MAN1 (LEMD3): 'rCrUrGrUrUrGrArUrArUrArUrArArArUrUrGrGrUrCrArGrUrCCA'
LAP2α (TMPO): 'rGrArUrArArArCrCrCrArGrArCrArArArGrArArArGrArUrArAAG'

RNAi to target LMNB1 was performed using Lipofectamine RNAiMAX reagent (Life Technologies, Cat. 13778030) and 10 nM human gene specific siRNA duplexes (abx903005, Abbexa Ltd) with following sequence:

LMNB1:'GCAGACUUACCAUGCCAAATTUUUGGCAUGGUAAGUCUGCTT'

As a control, a Trilencer-27 Universal Scrambled Negative Control siRNA Duplex (Origene) was used. Cells were visualized after 48 h post-transfection.

**Stable EMD-EGFP RPE1 cell line generation**. RPE1 cells were transfected with Emerin-pEGFP-C1. Forty-eight hours post-transfection cells were cultured in medium containing G-418 (400 μg/ml, Life Technologies, Cat. 11811-031) for 1 week. Afterward, single cells were plated onto 96-well plates. Colonies obtained from single cell expressing green fluorescence signal were selected and further propagated.

**Myoblast infection**. Ectopic expression of Emerin-EGFP in primary myoblasts was obtained by lentiviral transduction. Briefly, 293T cells were transfected with pLKO.1-EGFP-Emerin construct (containing pEGFP-emerin from Addgene, plasmid #61993 cloned into pLKO.1 backbone, 8566 bp) by calcium phosphate protocol (10 μg DNA for each 10-cm dish). Forty-eight hours later, viral supernatant was collected, 0.45-μm filtered and added to target cells. After 16 h of infection, medium was changed and myoblasts were seeded onto coverslips containing micro-patterns. After complete adhesion to fibronectin-coated linear patterns, cells were fixed in PFA 4% and used for successive immunofluorescent staining. For the analysis of the rescue experiments only cells expressing EMD-GFP were imaged.

**Boyden chamber chemotactic assay**. Chemotactic capacity of RPE-1 cells was monitored using 6-well Transwell permeable supports with 8 μm pores (Corning, Cat. 3428). Cells were detached by trypsinization, washed 2× in PBS, suspended at the concentration of $2 \times 10^5$/ml in DMEM (Lonza, Cat.BE12-614F) plus 0.1% FBS (Euroclone, Cat. ECS0181L) and seeded in the above inserts (final cell number: $3 \times 10^5$). Chemotactic migration was induced by adding DMEM (Lonza, Cat.BE12-614F) + 10% FBS (Euroclone, Cat. ECS0181L) + 50 ng/ml human recombinant epithelial growth factor (Vincil-Biochem, Cat. BPS-90201-3) in the bottom reservoir (2 ml of total volume). After 6 h of incubation at 37 °C, migrating cells were stained with NucBlue® Live Cell reagent (ThermoFisher, Cat. R37605). The cells on the upper side of the insert were scraped using a cotton swab. The cells present on the bottom side were imaged using widefield BX63 Olympus equipped with 4x objective. The total area of the well was scanned and the average number of cells per area was calculated.

**Western blot**. Western blot analysis was performed following standard procedures. Briefly, whole cell lysates were made in Laemmli buffer and the samples were resolved using Invitrogen Bolt 4–12% Bis-Tris gels (ThermoFisher Scientific).

Proteins were transferred to Amersham Protran nitrocellulose membrane using standard protocols and blocked in 5% milk in TBS-Tween (0.2%). Primary antibodies including anti-emerin (Monoclonal, NCL-EMERIN, mouse, Leica Biosystems, diluted at 1:5000), anti-lamin B1 (polyclonal, ab16048, rabbit, Abcam, diluted at 1:5000, anti-MAN1 (Polyclonal ab121854, rabbit, Abcam, diluted at 1:2500), anti-LAP2a (Polyclonal, ab5162, rabbit, Abcam diluted at 1:1000) anti-beta actin (Monoclonal, AC_15, ab6276, Abcam, diluted at 1:1000) were incubated at 4 °C overnight, followed by washing in TBS-T. The corresponding secondary antibodies conjugated to HRP (Horshradish Peroxidase) were incubated for 1 hour at room temperature. Blots were developed using Supersignal West Dura extended duration substrate (ThermoFisher Scientific Cat. 46641) using Chemidoc XRS + (BioRad).

**Confocal microscopy**. Images were taken every 0.3 μm of focal plane using z-stack function using oil immersion ×40 or ×60 objective (Leica Germany) with Nikon Eclipse Ti microscope (Nikon Instruments) equipped with the UltraVIEW VoX spinning-disc confocal unit (PerkinElmer) and Velocity software (PerkinElmer).

**Frequency distribution map preparation**. The images were processed using ImageJ software[58] with custom-built macro. After performing immunofluorescent staining, multiple confocal images were acquired. All the Z-stacks were projected by summing, images were rotated—Golgi signal was used to orient the cells in the same direction in relation to front-rear polarity. Then, images were registered to the reference nucleus using the turboreg ImageJ plugin[14]. The algorithm "rigid body" was set to find the combination of rotation and translation of the source to fit the target. The same roto-translation was applied to all channels. For each channel, all the registered images were combined in a single stack. Then, the stack was segmented using threshold value for each image in the stack using TopHat (value 20) filter. Finally, maps of distribution frequency were prepared by making a sum of all the images in the stack. While for some proteins a biased distribution was evident already at the single cell level, as shown in Supplementary Fig. 2a–e, for others it became evident only after averaging several single cell images.

**Centrosome-nucleus distance analysis and quantification**. Immunofluorescent staining using anti-pericentrin antibody was performed and nuclei were visualized using DAPI. After projecting Z-stacks, nuclei were registered to the reference nucleus using the turboreg ImageJ plugin. For each channel all the registered images were combined in a single stack. Then, the stack was segmented using threshold value for each image in the stack, using Yen (value 25) for pericentrin signal and Otsu (value 100) for nucleus. Finally, the custom-built ImageJ macro measured distance between the centrosome and the closest border of the nucleus for each cell. Value of 0 means that centrosome was positioned above the nucleus.

**Time-lapse microscopy for migration analysis**. RPE1 cells were plated on micro-patterned lines 16 h before imaging. In order to track the cells, nuclei were stained with NucBlue live stain (Thermo Fisher Scientific, Cat. R37605). Cells were imaged using a humidity- and temperature-controlled inverted wide-field microscope within an environmental chamber (Olympus). Cells were transferred to a live cell imaging workstation composed by an Olympus IX81 inverted microscope equipped with motorized stage and a Hamamatsu ORCA-Flash 4.0 camera. The images were collected every 20 min for a total recording time of 48–72 h using a PlanFL 4×/0.13 or a UPlanFLN ×10/0.30 phase-contrast dry objective with CellSens software (Olympus). Nuclei images were segmented and geometric centers were tracked with a global minimization algorithm. For this purpose, specific software in C++ with the OpenCV [http://opencv.willowgarage.com/wiki/] and the GSL [http://www.gnu.org/software/gsl/] libraries was developed. The migration analysis was performed by the C++ software coupled with R [www.R-project.org][11].

**Traction force microscopy**. Red fluorescent quantum dots (QDs) were deposited on silicone samples into confocal monocrystalline triangular arrays using electrohydrodynamic nanodrip printing[39]. Here, droplets of colloidal QD ink were ejected from a micro nozzle in an electric field. In this printing process, the highly regular arrays of QDs were produced by modulating both the electric field and nozzle position. Human hTERT RPE-1 cells were seeded on the fibronectin-coated silicone samples containing QDs and incubated for 16 h. Afterward, cells were fixed with 4% PFA and stained with DAPI and FITC-phalloidin (Sigma-Aldrich, Cat. P5282). Confocal images were taken every 0.3 μm of focal plane using z-stack function using oil immersion 60x objective (Leica Germany) with Nikon Eclipse Ti microscope (Nikon Instruments). The mechanical properties of the underlying silicone CY52-276 (Dow Corning) with a mixing ratio of 10:9 (A:B) yielded an elastic modulus of 5.16 kPa. This allowed the reconstruction of the traction field from a single image of the displaced QDs using Cellogram[59]. Here, the displaced QDs were detected, the displacement field to the QDs' resting position was inferred, and the tractions were calculated using the known material properties.

**Focal adhesion analysis and quantification**. In order to visualize focal adhesions, cells were stained with anti-paxillin antibody. Using the custom-built ImageJ[58] macro, the best z-plane was selected, the signal was segmented using Intermodes

(value 25) and the size of the particles was measured. For each cell, an average area of focal adhesions was calculated.

**Time-lapse microscopy and emerin dynamics analysis**. RPE1 cells stably expressing EMD-EGFP were plated on micro-patterned lines 16 h before imaging, nuclei were stained with NucBlue live stain (Thermo Fisher Scientific, Cat. R37605). Samples were imaged with inverted microscope Nikon ECLIPSE Ti2-E equipped with a motorized stage, a Photometrics Prime 95B sCMOS Camera and a 40x objective (Planfluor ELWD N.A. 0.60). Timepoints were collected every 10 min. for 44–48 h. The emerin enrichment at the NE was analyzed with a custom macro for ImageJ[14,58,60]. For the analysis, we selected cells with evident emerin bias. Thanks to the specific staining, nuclei were automatically segmented in all frames to identify both the center of the nucleus and the direction of motion. In the EMD-EGFP channel background was subtracted and at each time point the EMD signal intensity was measured along 2 μm inwards the nucleus edge. The front of the nucleus was defined as the interval of 45 degree in the direction of motion and colored in yellow. The back was defined as the 45-degree interval along the opposite direction and colored in red. The rest of the nucleus edge was colored in blue. The normalized intensity profile along the nuclear edge in function of the angle respect the direction of motion was finally plotted with R and ggplot2[60,61].

**Transmission electron microscopy**. Electron microscopic examination and immune EM gold-labeling based on pre-embedding, were performed as previously,[62]., a detailed description is explained below. EM based Morphological analysis: RPE1 cells grown on MatTek glass-bottom dishes dishes (MatTek Corporation, Cat. P35G-1.5-14-C) for 16 h. Samples were fixed with of 4% paraformaldehyde and 2,5% glutaraldehyde (EMS) mixture in 0.2 M sodium cacodylate pH 7.2 for 2 h at RT, followed by 6 washes in 0.2 sodium cacodylate pH 7.2 at RT. Then cells were incubated in 1:1 mixture of 2% osmium tetraoxide and 3% potassium ferrocyanide for 1 h at RT followed by 6 times rinsing in cacodylate buffer. Then the samples were sequentially treated with 0.3% Thiocarbohydrazide in 0.2 M cacodylate buffer for 10 min and 1% OsO4 in 0.2 M cacodylate buffer (pH 6,9) for 30 min. Then, samples were rinsed with 0.1 M sodium cacodylate (pH 6.9) buffer until all traces of the yellow osmium fixative have been removed, washed in de-ionized water, treated with 1% uranyl acetate in water for 1 h and washed in water again. The samples were subsequently embedded in Epoxy resin at RT and polymerized for at least 72 h in a 60 °C oven. Embedded samples were then sectioned with diamond knife (Diatome) using Leica ultramicrotome. Nano-gold labeling: RPE1 cells were grown on MatTek glass-bottom dishes (MatTek Corporation, Cat. P35G-1.5-14-C) for 16 h Afterward, specimens were fixed with a mixture of 4% paraformaldehyde and 0.05% glutaraldehyde in 0.15 M Hepes for 5 min at RT and then replaced with 4% paraformaldehyde in 0.15 M Hepes for 30 min RT followed by washing 6 times in PBS at RT and incubation with blocking solution for 30 min at RT. Then, cells were incubated with primary antibodies – anti-emerin (Leica Biosystems, NCL-EMERIN) and anti-nesprin-1 (Thermo Scientific, MA5-18077) diluted in blocking solution overnight at 4 °C. On the following day, the cells were washed 6 times with PBS at RT and incubated with goat anti-rabbit Fab' fragments coupled to 1.4 nm gold particles (diluted in blocking solution 1:100) for 2 h and washed 6 times with PBS at RT. Meanwhile, the activated GoldEnhanceTM-EM (Nanoprobes, Cat. 2113) was prepared according to the manufacturer's instructions and 100 μl were added into each well. The reaction was monitored by a conventional light microscope and was stopped after 5–10 min when the cells had turned "dark enough" by washing several times with PBS. Osmification followed: the cells were incubated for 1 h at RT with a 1:1 mixture of 2% osmium tetraoxide in distilled water and 3% potassium ferrocyanide in 0.2 M sodium cacodylate pH 7.4 and then rinsed 6 times with PBS and then with distilled water. The samples were dehydrated in 50% ethanol; 3 × 10 min in 70% ethanol; 3 × 10 min in 90% ethanol; 3 × 10 min in 100% ethanol. The samples were subsequently incubated for 2 h in 1:1 mixture of 100% ethanol and Epoxy resin (Epon) at RT, the mixture was then removed with a pipette and finally samples were embedded for 2 h in Epoxy resin at RT. The resin was polymerized for at least 10 h at 60 °C in an oven[63]. Sections were analyzed with a Tecnai 20 High Voltage EM (FEI) operating at 200 kV.

**Proximity ligation assay and quantification**. Proximity ligation assay was performed using commercially available Duolink PLA kit (Sigma-Aldrich, Cat. DUO92101-1KT). After 16 h incubation of hTERT RPE-1 cells on micro-patterned lines, they were fixed with 4% PFA and permeabilized with 0.1%Triton-X. Proximity ligation assay was performed according to the manufacturer's protocol. For this experiment, pairs of mouse anti-emerin (Leica Biosystems, NCL-EMERIN)/ rabbit anti-Lamin B1 (Abcam, ab16048) or mouse anti-emerin (Leica Biosystems, NCL-EMERIN)/rabbit anti-nesprin-1 (Sigma-Aldrich, HPA019113) antibodies were used. The nuclei were stained with DAPI and F-actin was visualized using FITC-phalloidin (Sigma-Aldrich, Cat. P5282). The images were projected and signals were quantified using the custom-built ImageJ macro using Triangle segmentation method. The DAPI signal was used to exclude the nuclear signal whereas the F-actin marked the area of the cell. For each cell, a number of signals in the cytoplasm was calculated.

**Drug treatment**. Cells were left for 16 h to attach on micro-patterned lines. Then, the drugs at the following concentrations and incubation time were added: Cytochalasin D (Sigma-Aldrich, Cat. C2618): 500 nM for 30 min., Nocodazole (Sigma-Aldrich, Cat. M1404): 1 μg/ml for 30 min., Blebbistatin (Merk, Cat. B0560): 25 μM for 15 min., Calyculin A (Abcam, Cat. ab141784): 50 nM for 15 min. Afterward, the cells were fixed with 4%PFA/PBS and immunofluorescence was performed.

**Statistics and reproducibility**. At least 10 single cells were randomly selected for each experiment (preferably >30 cells/experiment that were ten pooled to obtain a final map). For each map cells from at least 3 independent experiments were analyzed. For each distribution map an exact number of cells is stated in the figure. The distribution maps were tested using R studio (http://www.r-project.org/)[61]. The two-sided Kolmogorov–Smirnov test was used to compare protein distribution with uniform distribution (asterisk in black). The distribution of proteins was compared using the two-sided Cramér–von Mises. Asterisk in magenta were used for MOCK vs. EMDRNAi/Normal myoblasts vs. EDMD myoblasts, asterisk in blue for MOCK vs. drug-treated or other protein knock-down or asterisk in green for EDMD myoblasts vs. EDMD + EMD-EGFP myoblasts. The GraphPad Prism software was used to produce graphs (7.0 d licensed for IFOM) and statistical analysis for Nuc-MTOC distance, Traction, PLA quantification, FA analysis, migration. The two-sided Kolmogorov–Smirnov or Kruskal–Wallis tests (confidence interval = 95%) were used to determine the significance between two or three groups. For chemotaxis assay a two-tailed Wilcoxon matched-pair test (confidence interval = 95%) was used to compare the conditions. The corresponding P values of the tests are shown in the figure.

**Stochastic simulation**. A one-dimensional lattice was used to mimic the ER. As schematically represented in Fig. 2g, we divided the lattice in three segments: the front ($L_1$), the nucleus ($L_N$) and the back ($L_2$). The NE was defined as the single voxels between the front ($NE_F$) and the nucleus or the back ($NE_B$) and the nucleus respectively. The time step of the simulation, $\Delta t$, was fixed to 1 s, then the dimension of the voxels of the lattice, $x$, was computed accordingly to the relation: $\Delta t = x^2/D$, where D is the diffusion coefficient. We assumed the diffusion coefficient of EMD in the ER to be similar to the one of GFP: 10 μm$^2$/s[64]. The number of voxels in the lattice was determined in order to have the lattice length equal to the typical experimental one: 80 μm. We assumed that in total in the ER is arriving 1 EMD molecule per second and that the mean-life of the protein is 50h[65].We used these values to compute the probability per voxel and per unit of time (Kon) that a new molecule appears in the ER and the probability per unit of time (Koff) that a molecule is degraded. Molecules in the front and back could freely move but, because EMD interacts whit other elements of the NE that stabilized it, we roughly assumed the diffusion coefficient of EMD at the NE neglectable. We started the simulation with a number of EMD molecules at the steady-state and uniformly distributed in the ER, then molecules were allowed to move. At each time step new molecules were generated at both sides of the ER with equal probability Kon and in the ER, as well as at the NE, with probability Koff, molecules were degraded. In Supplementary Movie 2 we show the temporal evolution of the simulation with proportion between $L_2$ and $L_1$ similar to the experimental data: $L_2 = L_{1+}L_N$. In Fig. 2g, we show the results of different simulations and we plotted the ratio between the number of EMD molecules at the $NE_F$ and $NE_B$ in function of different values of $L_2/(L_1 + L_N)$ and we compared it with a mathematical model. Our simulations support the idea that an asymmetric distribution of ER in the front versus the back of the cell is sufficient to generate an asymmetric distribution of elements at the NE. A detailed mathematical model is described in Supplementary Text.

**Reporting summary**. Further information on research design is available in the Nature Research Reporting Summary linked to this article.

## Data availability
The authors declare that the data supporting the findings of this study are available in Supplementary Information files or from the corresponding author upon reasonable request. The source data underlying Figs. 2d, 4b, 4d, 5c and Supplementary Figs. 2a, 2i, 3c, 4c, 4g, 5a, 5b, 6a, 6b, 6c, 6f, 7a, 7f are provided as Source Data file.

## Code availability
Custom-built ImageJ macros are available from the corresponding author upon request.

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

## Acknowledgements

This work was supported by Italian Association for Cancer Research (AIRC) fellowships for Paulina Nastały (Project code: 19416) and Gururaj Rao Kidiyoor (Project code: 19464) This project was founded by Fondazione Umberto Veronesi Post-doctoral fellowships for Paulina Nastały. The authors would like to thank IFOM Imaging Facility and IFOM Cell Biology Unit for help with performing experiments. We would like to acknowledge all the participants to the Milano 4D-Nucleus meetings, Giorgio Scita, Nils Gauthier, Matthieu Piel, and Kristina Havas Cavalletti for helpful comments and discussions. Martina Galli and Ylli Doksani for generation of pLKO-EMD-GFP plasmid. Bas van Steensel, Marloes van der Zwalm, and Daniel Peric Hupkes for sharing with us the DamID plasmids. We would like to acknowledge Shivashankar GV for DN-KASH-EGFP plasmid. We thank Dr. Marina Mora from the "C.Besta" IRCCS Neurological Institute and Telethon Biobank for the primary myoblasts samples.

## Author contributions

P.M. designed experiments and supervised the study. P.N. designed, carried out and interpreted experiments. Divya.P. and A.P. performed biochemical experiments. G.R.K. performed live cell imaging with EMD-EGFP cells. S.M. performed chemotaxis and myoblast infection experiment. T.L, D.P., and A.F supported this project with TFM and analysis. G.V.B. and A.M. performed TEM. O.M.R. and M.C.L developed the mathematical model. S.L. helped with generation of stable cell line P.M. and P.N. carried out data and statistical analysis. P.M. and P.N. wrote the paper.

## Competing interests

The authors declare no competing interests.
