## [Peer Review File · Nature Communications]

Reviewers' comments:

Reviewer #1 (Remarks to the Author):

Summary:

Nastaly et al. study the role of emerin in nuclear and cell polarity. Emerin depleted cells exhibited loss in polarity and decreased cell migration. The distribution of emerin was modeled mathematically. Finally, replenishment of emerin in myoblasts from EDMD patients led to establishment of polarity in these cells.

Positives:

This work finds a new role for emerin in establishing cell and nuclear polarity and the distribution of emerin, modeled mathematically, provides a possible explanation for the biased distribution of other nuclear envelope molecules.

Critiques:

- 1) An experimental validation of the possible explanation of establishment of polarity, which was modeled mathematically, is lacking. It remains unclear to this reader how emerin causes nuclear polarization. Experimentally, the role of emerin in nuclear polarization is purely correlative.
- 2) There's not enough novelty to warrant publication in NComms. Two papers have already shown that the nucleus can be polarized, in this case apically. Hence, the novelty in this paper is somewhat limited.
- 3) While the authors thoroughly assess the distribution of many molecules in the nucleus following depletion of emerin, this deep phenotypic assessment remains limited to just one cell manipulation (i.e. just emerin KD). Once cell manipulation is not nearly sufficient to establish a molecular mechanism for nuclear polarization. For instance, do the knockdown of LAP2 α and MAN1 affect nuclear polarity?

Reviewer #2 (Remarks to the Author):

In this manuscript "Front-rear polarity of the nucleus" Nastaly et al. demonstrate that the nucleus of the cell polarizes along the front-rear axis of the cells on micropatterned lines and that this polarization is dependent on the protein Emerin (EMD). The authors hypothesize that cytoskeletal polarity is transmitted to the nucleus to induce polarization in this organelle. They demonstrate that in the absence of Emerin, nuclear proteins (e.g. lamins, and histone modification marks) are no longer polarized. Overall, this research is very exciting as it demonstrates that the nucleus exhibits polarity similar to the bulk cytoplasm and begins to explain how this polarity is established. However, this manuscript is missing introduction and discussion sections. Furthermore, I would recommend several additional experiments prior to strengthen the conclusions before it would be ready for publication.

Major comments and concerns:

1. This manuscript is currently lacking an introduction and discussion as would be expected for a full submission. These sections are required to fully appreciate the novel work demonstrated and place it in the proper context. It would be beneficial to the reader if the introduction described what has been shown previously, and specifically addressed the potential ramifications of nuclear polarity. For example, Emerin has previously been shown to be involved in nuclear mechanotransduction (Lammerding J et al. JCB 2005), but this and other studies are not discussed at all.

Do the authors hypothesize that the nuclear polarity itself is necessary for normal cell behavior, perhaps through effects in gene expression, or is it a byproduct of the cytoskeletal/organelle polarity observed in the bulk cytoplasm? What if any are the functional consequences of disrupting the observed polarity?

2. It would be helpful to include an additional figure to demonstrate the workflow for the nuclear distribution maps. This figure could be either added to Figure 1 or made as a supplemental figure. Because the bulk of the conclusions depend on this analysis, it would be helpful to the readers to readily comprehend how the authors are averaging the nuclear data in order to generate the distribution maps. How are different nuclear sizes or shapes accounted for? If this method were clearer the conclusions would be significantly strengthened.

3. In Figure 2b, the authors demonstrate that EMD knockdown dramatically decreases cellular traction forces. Do other perturbations of cytoskeletal tractions (e.g. blebbistatin, actin targeting drugs, or perturbations to Rho GTPase) likewise perturb the organization of nuclear proteins like EMD and others? Does increasing contractility increase nuclear polarization? These experiments are essential to delineate whether the EMD knockdown is altering nuclear protein localization via a structural interaction with the nuclear envelope or ER, or if it is simply a by-product of reducing polarized cellular tractions upon EMD knockdown. Do these traction perturbations effect ER localization and nuclear polarity independently of EMD manipulations?

3. In lines 107-114, the authors seem to suggest that sites of active transcription become more localized to the rear tip of the nucleus upon EMD knockdown, noting trends in nuclear actin and Ser5 phosphorylated RNAPII follow this trend. However, these observations appear to be in conflict with a concurrent trend of H3K27me3, an epigenetic mark of heterochromatin, as well localizing to the rear tip of the nucleus upon EMD knockdown. Additional evidence in support of a re-localization of active transcription sites is lacking. If the authors want to suggest that sites of active transcription become re-localized upon EMD knockdown, this should be directly measured through 5-Ethynyl Uridine labeling or other comparable methods.

4. In lines 112-113, the authors cite localization of H3K9ac and H3K4me3 in support of similarly distributed Ser5 phosphorylated RNAPII. However, the authors do not appear to include the data regarding localization of these histone marks in EMD knockdown cells. These experiments should be included.

5. The data pertaining to the chromosome territories do not appear to support the conclusions presented in lines 121-128. For example, in lines 125-126, the authors state: "In our analysis, each CT showed a unique map of preferential distribution in relation to cell polarity". This wording grossly overstates the data presented in the associated figures. In fact, much of the chromosome territory data does not appear to be statistically significant in patterned and non-patterned cells. Furthermore, of the six chromosome territories measured, only one appears to significantly re-localize upon EMD knockdown. I suggest re-wording this paragraph in order to more accurately explain the data.

Minor comments and concerns:

1. In the western blot in figure S6-A, Lamin-B1 appears to be nearly depleted. (This is especially noticeable when compared to the western blot presented in figure 2H.) As such, I am concerned that the localization data presented in S6C could be spuriously attributed to the normalization process. Please include representative images of the Lamin-B1 antibody staining in this or a supplementary figure.

2. Figure 2C is nearly indistinguishable. The labeling density of the immunogold is so low that it's very challenging to see protein localization. How can the authors be sure this is specific labeling? A non-specific control would be helpful as both nesprin and EMD seem to be randomly distributed throughout the cytoplasm and nucleus. It would also be helpful in the figure to label where the nuclei vs. cytoplasm are.

3. In Figure 2D in which the authors demonstrate the proximity labeling results, it is very difficult to observe the increased cytoplasmic fraction of EMD. It would perhaps be helpful to invert the image or show a higher magnification image to more clearly demonstrate the data.

4. Is DAPI staining uniform or does it also display polarized intensity differences?

Reviewer #3 (Remarks to the Author):

In this paper, the authors demonstrate that the nuclear envelope proteins and chromatin exhibit polarized distribution spatially. The findings suggest that emerin, a protein found in the nuclear envelope and recruited from the endoplasmic reticulum, plays a critical role in generating polarity of proteins.

While the finding is novel, I have the following reservations/concerns about the evidence provided in the paper:

1. Fig. 2b shows evolution of the emerin intensity. If I look at the initial plot (0 min) and the final plot (200 min), all I gather is that the 0 min plot is essentially translated vertically up. This implies that the intensity has gone up uniformly in all the domains of the NE. It does not show that the intensity preferentially increases at the poles with increasing time. This brings me to my key concern. The paper does not present strong evidence that the nuclear polarity develops when the cell becomes polarized. Data that clearly shows development of polarity in a time-dependent manner from a non-polar state is needed to appreciate the claim made by the authors.

2. The predictions from the computational study performed to support the idea that the EMD polarity develops from an asymmetric size distribution of ER at the two poles appear almost obvious. The assumptions are built into the model, and hence the model is not likely to lead to any other outcome.

3. The key proteins that show reduction in front polarity are LMN B1, LMN A/C and RNAPII Ser5. SUN2 shows an opposite trend and develops frontal polarity in EMD knock-down cells. The other proteins do not show a reduction in intensity at the front pole. Some do show increase in intensity in other domains. For example, Chr3, nesprin 1 and nesprin 2 show no reduction in the intensity at the frontal pole in the EMD knock-out cells. This dampens the enthusiasm for the main claim of the study. The authors should limit the generality of their claim to match with the data.

4. How is the opposite trend of SUN2 explained? Can the mathematical model be used to gain some insights?

5. Did the authors knock-down any other NE protein and measured the impact on frontal polarity of the NE proteins? Could any other NE protein potentially have a similar impact on regulating the NE polarity?

Overall, the findings, if substantiated further, are novel and will present a new perspective in the field. Hence, the paper would be of considerable interest to the community.

Point-by-point answer to reviewer's comments to manuscript "Front-rear polarity of the cell nucleus" (NCOMMS-19-22054A). :

Reviewer #1 (Remarks to the Author):

Summary:

Nastaly et al. study the role of emerin in nuclear and cell polarity. Emerin depleted cells exhibited loss in polarity and decreased cell migration. The distribution of emerin was modeled mathematically. Finally, replenishment of emerin in myoblasts from EDMD patients led to establishment of polarity in these cells.

Positives:

This work finds a new role for emerin in establishing cell and nuclear polarity and the distribution of emerin, modeled mathematically, provides a possible explanation for the biased distribution of other nuclear envelope molecules.

Critiques:

1) An experimental validation of the possible explanation of establishment of polarity, which was modeled mathematically, is lacking. It remains unclear to this reader how emerin causes nuclear polarization. Experimentally, the role of emerin in nuclear polarization is purely correlative.

We regret that our message was not sufficiently clear. By adding results and discussion sections, we have aimed to improve the clarity of our manuscript. We would like to stress that it was never our claim that emerin causes nuclear polarity. Indeed, nuclear polarity parallels front-rear cell polarity and emerin is one of the proteins at the nuclear envelope that we found to be asymmetrically distributed. Moreover, we show that the observed biased distributions of some proteins of the nuclear envelope, or inside the nucleus, can depend on emerin. However, this is not true for all the elements that we tested, pointing to the conclusion that clearly emerin plays an important role in polarity transmission from the cytoplasm to the nucleus, but it is not the only player.

2) There's not enough novelty to warrant publication in NComms. Two papers have already shown that the nucleus can be polarized, in this case apically. Hence, the novelty in this paper is somewhat limited.

We agree with the reviewer that these two seminal publications, cited in our original manuscript, show that the nucleus can be polarized. However, not only we investigate the front-rear polarity, which is a different phenomenon than the apico-basal one, but additionally, by systematically and quantitatively analyzing spatial distribution of different components of the nuclear envelope, nucleoplasm and chromatin, we reveal a completely novel perspective of cellular organization. It is also important to notice that front-rear polarity is an intrinsic property of all adherent cells, which are almost always able to autonomously establish an axis of polarity and migrate¹. This increases the general message and impact of our findings.

3) While the authors thoroughly assess the distribution of many molecules in the nucleus following depletion of emerin, this deep phenotypic assessment remains limited to just one cell manipulation (i.e. just emerin KD). Once cell manipulation is not nearly sufficient to establish a molecular mechanism for nuclear polarization. For instance, do the knockdown of LAP2 α and MAN1 affect nuclear polarity?

We thank the reviewer for appreciating our effort to produce a precise and in-depth description of the spatial distributions of different components of the nuclear envelope and nucleoplasm in front-rear polarized cells. In the previous version of our manuscript we used two different cell types, the RPE1 cell line (CTRL or emerin knock-down), and primary

myoblasts from a healthy donor or from an emerin-deficient patient. Following the reviewer's suggestion, we investigate how different LEM domain-containing proteins can affect nuclear polarity. We therefore analyzed a LAP2 α and MAN1 knock-down. Additionally, we also show the results of a LMN B1 knock-down. These new experimental results are included in the revised version of our manuscript as "Effect of LAP2 α , MAN1 and LMN B1 knock-down also affect nuclear polarity" paragraph.

Reviewer #2 (Remarks to the Author):

In this manuscript “Front-rear polarity of the nucleus” Nastaly et al. demonstrate that the nucleus of the cell polarizes along the front-rear axis of the cells on micropatterned lines and that this polarization is dependent on the protein Emerin (EMD). The authors hypothesize that cytoskeletal polarity is transmitted to the nucleus to induce polarization in this organelle. They demonstrate that in the absence of Emerin, nuclear proteins (e.g. lamins, and histone modification marks) are no longer polarized. Overall, this research is very exciting as it demonstrates that the nucleus exhibits polarity similar to the bulk cytoplasm and begins to explain how this polarity is established. However, this manuscript is missing introduction and discussion sections. Furthermore, I would recommend several additional experiments prior to strengthen the conclusions before it would be ready for publication.

Major comments and concerns:

1. This manuscript is currently lacking an introduction and discussion as would be expected for a full submission. These sections are required to fully appreciate the novel work demonstrated and place it in the proper context. It would be beneficial to the reader if the introduction described what has been shown previously, and specifically addressed the potential ramifications of nuclear polarity. For example, Emerin has previously been shown to be involved in nuclear mechanotransduction (Lammerding J et al. JCB 2005), but this and other studies are not discussed at all.

We thank the reviewer for raising this point; we modified our manuscript accordingly. A regular introduction with a more careful consideration of the existing literature is now present in the manuscript.

Do the authors hypothesize that the nuclear polarity itself is necessary for normal cell behavior, perhaps through effects in gene expression, or is it a byproduct of the cytoskeletal/organelle polarity observed in the bulk cytoplasm? What if any are the functional consequences of disrupting the observed polarity?

This is a very interesting point. We hypothesize that nuclear polarity is a consequence of the asymmetric distribution of cellular organelles, mainly the ER. But this does not exclude that the asymmetry could itself be necessary for cell homeostasis, affecting distributions of nuclear components and/or driving specific gene expression programs. As we observed, emerin is partially responsible for nuclear polarity transmission. In human, the consequence of emerin loss is the Emery Dreifuss muscular dystrophy. While we cannot directly prove that the disease is caused by the loss of nuclear polarity, we show that in primary myoblasts from an EDMD patient nuclear polarity is perturbed and that the ectopic expression of emerin partially rescues this phenotype.

2. It would be helpful to include an additional figure to demonstrate the workflow for the nuclear distribution maps. This figure could be either added to Figure 1 or made as a supplemental figure. Because the bulk of the conclusions depend on this analysis, it would be helpful to the readers to readily comprehend how the authors are averaging the nuclear data in order to generate the distribution maps. How are different nuclear sizes or shapes accounted for? If this method were clearer the conclusions would be significantly strengthened.

We apologize to the reviewer for this lack of clarity. In the revised manuscript we add a separate figure (Fig. 1) that better explains the applied workflow. In the previous version we included also Supplementary Fig. 1 where we show some examples of distribution maps with or without orientation. We also rephrased and clarified this point in the paragraph entitled “Distribution map preparation”.

We do not modify the acquired images that are averaged. This means that we are not stretching the nucleus pictures to force them to have all the same size. We just orient the images using the Golgi localization as marker of directionality, and then we apply a rigid body

registration algorithm [<http://bigwww.epfl.ch/thevenaz/stackreg/>] to register the nuclei. All other channels are registered accordingly. To perform this analysis, we did not develop specific software, we simply exploited existing ImageJ plugins. The variation of nuclear size and shape can be appreciated looking at the color map of the nuclei. When we perform the statistical test we consider the intensity of signal along the front-rear axis and we normalize the signal for the average nucleus (0 is the front, 1 the rear).

3. In Figure 2b, the authors demonstrate that EMD knockdown dramatically decreases cellular traction forces. Do other perturbations of cytoskeletal tractions (e.g. blebbistatin, actin targeting drugs, or perturbations to Rho GTPase) likewise perturb the organization of nuclear proteins like EMD and others? Does increasing contractility increase nuclear polarization? These experiments are essential to delineate whether the EMD knockdown is altering nuclear protein localization via a structural interaction with the nuclear envelope or ER, or if it is simply a by-product of reducing polarized cellular tractions upon EMD knockdown. Do these traction perturbations effect ER localization and nuclear polarity independently of EMD manipulations?

We thank the reviewer for this question, and we agree that it is of particular interest to test how perturbing the cytoskeleton would affect nuclear polarity. As suggested, we treated RPE1 cells on micro-patterned lines with cytochalasin D, blebbistatin, calyculin A, and nocodazole. We observed that even if the nuclear and cell shape changed dramatically, the ER and Golgi frontal localization was preserved. None of the drugs abolished the emerin frontal enrichment, however, cytochalasin D and nocodazole significantly decreased it (Fig. 4f). On the other hand, calyculin A slightly increased the EMD frontal polarization. The data is presented in the “Nuclear polarity persists upon drug treatment” paragraph.

3. In lines 107-114, the authors seem to suggest that sites of active transcription become more localized to the rear tip of the nucleus upon EMD knockdown, noting trends in nuclear actin and Ser5 phosphorylated RNAPII follow this trend. However, these observations appear to be in conflict with a concurrent trend of H3K27me₃, an epigenetic mark of heterochromatin, as well localizing to the rear tip of the nucleus upon EMD knockdown. Additional evidence in support of a re-localization of active transcription sites is lacking. If the authors want to suggest that sites of active transcription become re-localized upon EMD knockdown, this should be directly measured through 5-Ethynyl Uridine labeling or other comparable methods.

We would like to thank the reviewer for this observation. Indeed, the spatial resolution of our maps, which corresponds to a probability distribution, is not sufficient to infer or exclude the co-localization of different chromatin domains. We agree that additional (and more demanding) experiment would be necessary to prove this point but we think that this is outside the main scope of our manuscript, then, as it was suggested by the Editorial Team, we toned down the findings about transcription.

We can add that emerin was previously reported by Le *et al.* (2016) to be involved in transcription alteration. Its force-driven enrichment at the outer nuclear membrane can cause a switch from H3K9me_{2,3} to H3K27me₃ occupancy at constitutive heterochromatin, resulting in attenuation of transcription and subsequent accumulation of H3K27me₃ at facultative heterochromatin².

4. In lines 112-113, the authors cite localization of H3K9ac and H3K4me₃ in support of similarly distributed Ser5 phosphorylated RNAPII. However, the authors do not appear to include the data regarding localization of these histone marks in EMD knockdown cells. These experiments should be included.

We thank the reviewer for spotting this absence and as suggested these data were included in Supplementary Fig. 2 k.

5. The data pertaining to the chromosome territories do not appear to support the conclusions

presented in lines 121-128. For example, in lines 125-126, the authors state: “In our analysis, each CT showed a unique map of preferential distribution in relation to cell polarity”. This wording grossly overstates the data presented in the associated figures. In fact, much of the chromosome territory data does not appear to be statistically significant in patterned and non-patterned cells. Furthermore, of the six chromosome territories measured, only one appears to significantly re-localize upon EMD knockdown. I suggest re-wording this paragraph in order to more accurately explain the data.

In the revised version of our manuscript we now state: “In our analysis, chromosomes 12, 18 and 22 were non-uniformly distributed (Supplementary Fig. 3a-c). Chromosome 3, as the only CT, lost its frontal enrichment upon EMD knock-down (Fig. 2i; Supplementary Fig. 2a, g)”. We also re-phrased the “Discussion” so it explains the data in a more accurate way.

Minor comments and concerns:

1. In the western blot in figure S6-A, Lamin-B1 appears to be nearly depleted. (This is especially noticeable when compared to the western blot presented in figure 2H.) As such, I am concerned that the localization data presented in S6C could be spuriously attributed to the normalization process. Please include representative images of the Lamin-B1 antibody staining in this or a supplementary figure.

The representative image of lamin B1 antibody staining for normal and EDMD myoblasts was included in Supplementary Fig. 7e.

2. Figure 2C is nearly indistinguishable. The labeling density of the immunogold is so low that it's very challenging to see protein localization. How can the authors be sure this is specific labeling? A non-specific control would be helpful as both nesprin and EMD seem to be randomly distributed throughout the cytoplasm and nucleus. It would also be helpful in the figure to label where the nuclei vs. cytoplasm are.

The possible reason of the invisibility of the gold particles was the very low magnification level of the image. In the new version of Figure 4c we increased the size of images and enlarged the area indicated with boxes. We labeled the compartments of interests: nucleus (Nuc), inner nuclear membrane (INM), outer nuclear membrane (ONM), endoplasmic reticulum (ER), actin filaments (Ac). Arrows indicate the gold particles present in selected compartments.

The specificity of labeling was proved using standard rule for the immune electron microscopy: the labeling densities of gold (the number of gold particles per 1 μm of membrane length) over ONM and INM was more than 4-fold higher than that over the outer mitochondrial membrane or over Golgi membrane where emerin was not present. For nesprin-1 we calculated the ratio between the labeling density over actin filaments versus such density over the mitochondrial matrix. This ratio was 5.1 ± 0.5 . This suggests that the labeling was specific. Additionally, we always use the standard method to determine the specificity, namely, we eliminate the primary antibodies or replaced them with irrelevant antibodies. In all of these cases the labeling was completely absent (not shown) because we used the very strong blocking solution. Moreover, below we present several images of cryo-sections where the labeling for nesprin-1 was very clear due to higher level of the antibody dilution. Gold particles, nuclei and the ER are shown with arrows in all images.

Fig. I Cellular localization of nesprin-1 and emerin. (A-C) Labelling for nesprin-1 in cryosections. (B) Enlargement of the area inside the white box form. (D-E) Epon sections. Pre-embedding. Nano-gold labelling for emerin with subsequent gold enhancement. Nano-gold enhancement. N – nucleus. White arrows show gold labelling over nuclear envelope. Black arrow shows gold labelling over the endoplasmic reticulum. Scale bars are shown on the bottoms of images.

3. In Figure 2D in which the authors demonstrate the proximity labeling results, it is very difficult to observe the increased cytoplasmic fraction of EMD. It would perhaps be helpful to invert the image or show a higher magnification image to more clearly demonstrate the data.

The zoom to the cytoplasmic region of the cell was added to the Fig. 4d.

4. Is DAPI staining uniform or does it also display polarized intensity differences?

In our analysis the averaged DAPI staining signal is always uniform.

Reviewer #3 (Remarks to the Author):

In this paper, the authors demonstrate that the nuclear envelope proteins and chromatin exhibit polarized distribution spatially. The findings suggest that emerin, a protein found in the nuclear envelope and recruited from the endoplasmic reticulum, plays a critical role in generating polarity of proteins.

While the finding is novel, I have the following reservations/concerns about the evidence provided in the paper:

1. Fig. 2b shows evolution of the emerin intensity. If I look at the initial plot (0 min) and the final plot (200 min), all I gather is that the 0 min plot is essentially translated vertically up. This implies that the intensity has gone up uniformly in all the domains of the NE. It does not show that the intensity preferentially increases at the poles with increasing time. This brings me to my key concern. The paper does not present strong evidence that the nuclear polarity develops when the cell becomes polarized. Data that clearly shows development of polarity in a time-dependent manner from a non-polar state is needed to appreciate the claim made by the authors.

We really apologize to the reviewer for the lack of clarity but in that panel we intend to show that also in living cells, by ectopically expressing an emerin-EGFP construct, it is possible to observe an asymmetric distribution of signal at the nuclear envelope. The different plots refer to the intensity profiles of emerin in the same cell (the one shown on the left) migrating on a fibronectin-coated micro-patterned line and observed at different time point (from 0 to 200 min). Please refer also to the Supplementary Movie 1. What we consider interesting to notice in this figure are the profile fluctuations in time that it is impossible to appreciate in our other experiments, performed in fixed samples averaging many cells. We have now changed the description of the figure.

Moreover, planar polarity is a natural/intrinsic state of the cell. Almost all adherent cells, when plated on a 2D surface, after spreading, self polarize and start to move on a random direction. The micro-patterned lines that we are using in our experiments are not triggering a polarized state, as it is possible to appreciate also looking the 2D distribution maps presented in supplementary figure 4.

To force RPE1 cells in a non-polarized state we plated them on a glass coated with poly-D-lysine and PLL-g-PEG (15 $\mu\text{g/ml}$ and 0.5 mg/ml , mixed 1:1), that triggers cell adhesion only by difference of charge between the surface and the cell. In this non-physiological condition RPE1 cells acquire atypical cell and nuclear shape (Fig. IIa) and they reduce their speed by $\sim 50\%$ (Fig. IIb). When we perform our analysis in this condition we found that at a non-polarized localization of the Golgi corresponds to a non-polarized distribution of emerin at the nuclear envelope (Fig. IIc).

Fig. II Induction of a non-polarized state in RPE-1 cells. **a** Representative cells on poly-D-lysine/PLL-g-PEG-coated glass (15 $\mu\text{g}/\text{ml}$ and 0.5 mg/ml , mixed 1:1), and fibronectin-coated glass (25 $\mu\text{g}/\text{ml}$). **b** Quantification of the speed of cells on poly-D-lysine/PLL-g-PEG-coated glass (15 $\mu\text{g}/\text{ml}$ and 0.5 mg/ml , mixed 1:1), and fibronectin-coated glass (25 $\mu\text{g}/\text{ml}$) $n_{\text{fibronectin}}=565$ cells, $n_{\text{PLL-g-PEG/Poly-D-lysine}}=449$ cells. **c** single cell (left panel) and distribution maps for the nucleus, Golgi and emerlin in cells plated on poly-D-lysine/PLL-g-PEG-coated glass (15 $\mu\text{g}/\text{ml}$ and 0.5 mg/ml , mixed 1:1).

2. The predictions from the computational study performed to support the idea that the EMD polarity develops from an asymmetric size distribution of ER at the two poles appear almost obvious. The assumptions are built into the model, and hence the model is not likely to lead to any other outcome.

We now have done our best to clarify this point in the text; in Supplementary Information we also included a Supplementary Information Fig. S1 illustrating the outcome of some variations of the proposed model. The model has a descriptive intent and is built on simple assumptions: (i) emerlin synthesis is proportional to ER available surface, (ii) emerlin motion in the ER is diffusive, and (iii) diffusion in the nuclear envelope is slow. The outcome of the model (asymmetry in the emerlin repartition with respect to opposite sides of the nucleus in proportion to the amount of ER present at either side) is a non-trivial consequence of these assumptions, and the interest relies in the fact that this is the minimal set of assumptions yielding to an outcome for emerlin asymmetry in line with the experimental observations. Indeed, variations in the model ingredients (see Supplementary Information Fig. S1) lead to different predictions. For example, the expected emerlin gradient would be opposite if emerlin diffusion coefficient in the ER were of several orders of magnitude less, or would be absent if emerlin synthesis were not proportional to the ER surface. It is important to note that what drives diffusion is concentration gradients, hence a biased ER would not be sufficient in absence of a process biasing the concentration (key assumption (i) in our model).

3. The key proteins that show reduction in front polarity are LMN B1, LMN A/C and RNAPII Ser5. SUN2 shows an opposite trend and develops frontal polarity in EMD knock-down cells. The other proteins do not show a reduction in intensity at the front pole. Some do show increase in intensity in other domains. For example, Chr3, nesprin 1 and nesprin 2 show no reduction in the intensity at the frontal pole in the EMD knock-out cells. This dampens the enthusiasm for the main claim of the study. The authors should limit the generality of their claim to match with the data.

We thank the reviewer for this comment, and we agree. Emerin is clearly not the only responsible protein for nuclear polarity transmission. As we show it has an important role but is not the only player.

4. How is the opposite trend of SUN2 explained? Can the mathematical model be used to gain some insights?

SUN2 was shown to be present in linear arrays harnessing the nuclear envelope to retrograde actin flow³. In the control single cell we can observe that it decorates the upper nuclear envelope, possibly where such linear arrays form (Supplementary Fig. 2d), which is lost in EMD knock-down. Emerin was shown to ensure the directionality of actin cable flow and its knock-down was shown to cause abnormal transmembrane actin-associated nuclear line coupling to the nucleus⁴. We can speculate that changes in actin flow induced by emerin depletion, altered SUN2 coupling to transmembrane actin-associated nuclear lines.

Moreover, our model cannot describe the dynamics of SUN2 together with that of EMD, especially since there might exist a competition (for the same binding sites on the nuclear membrane) between the two proteins. Indeed, in absence of emerin, SUN2 presents EMD-like front/back asymmetry, while its concentration appears to be unbiased in presence of EMD. The implementation of such a competitive dynamics would require supplementary hypothesis and equations and a more mechanistically detailed modeling, which was beyond our scopes in this work.

5. Did the authors knock-down any other NE protein and measured the impact on frontal polarity of the NE proteins? Could any other NE protein potentially have a similar impact on regulating the NE polarity?

We appreciate this comment. In the revised version of our manuscript we investigate how different LEM domain-containing proteins affects nuclear polarity. We then performed LAP2 α and MAN1 knock-down. Additionally, we also study LMN B1 knock-down. These new results are included in the revised version of our manuscript as “Effect of LAP2 α , MAN1 and LMN B1 knock-down also affect nuclear polarity” paragraph.

Overall, the findings, if substantiated further, are novel and will present a new perspective in the field. Hence, the paper would be of considerable interest to the community.

We really appreciate this positive comment.

References

1. Maiuri, P. *et al.* NIH Public Access. **22**, (2013).
2. Le, H. Q. *et al.* Mechanical regulation of transcription controls Polycomb-mediated gene silencing during lineage commitment. *Nat. Cell Biol.* **18**, 864–875 (2016).
3. Luxton, G. W. G., Gomes, E. R., Folker, E. S., Vintinner, E. & Gundersen, G. G. Linear arrays of nuclear envelope proteins harness retrograde actin flow for nuclear movement. *Science (80-.)*. **329**, 956–959 (2010).
4. Chang, W., Folker, E. S., Worman, H. J. & Gundersen, G. G. Emerin organizes actin flow for nuclear movement and centrosome orientation in migrating fibroblasts. *Mol. Biol. Cell* **24**, 3869–3880 (2013).

Reviewers' comments:

Reviewer #1 (Remarks to the Author):

The authors addressed my criticisms comprehensively and added two additional cell manipulations to their assessment of nuclear polarity. I recommend publication in Nature Comm.

Reviewer #2 (Remarks to the Author):

In the revised manuscript "Front-rear polarity of the cell nucleus", the authors have added additional discussion and experimental data to address the my, and the other reviewer concerns. Although I believe that the results are interesting in their own right, I still have substantial concerns regarding the formatting and conclusions of the manuscript. There is an impressive amount of work presented here; however, the presentation of the data and the biological conclusions are lacking interpretation and support. My comments on the previous submission indicated this (e.g. a complete lack of introduction section) and the experimental suggestions were intended primarily to try and elucidate some of the mechanism behind the results presented (e.g. is this mediated by forces, cytoskeleton, or simply organelle location?). Testing this would require manipulating both forces, cytoskeletal, and organelle locations experimentally. Unfortunately, it appears that the results from these experiments were largely inconclusive. Or at least, are presented in such a way, as that I'm unable to determine the underlying biology driving the nuclear polarization. Further, the mathematical model presented is not experimentally tested in any substantive way.

Much of the manuscript is devoted to describing the experiments and results, but it is often left to the reader make sense of the significance and guess about the biological mechanism. Seemingly contradictory results are glossed over as being "interesting", but not explored further. As a result, the major conclusions that I can draw from the manuscript are that in polarized cells, some nuclear proteins also become polarized and that EMD knock down affects some, but not all of these proteins and sometimes does so in opposing ways. The proteins that polarize and the susceptibility of these proteins to EMD knockdown are different in different cells. I feel like there is the potential for an interesting story to be made from the work presented here, but as it stands, the experiments performed, and the description of the results, is not sufficient to warrant publication in Nature Communications.

Major comments and concerns:

1) While I appreciate that the authors have reformatted the manuscript to be more easily understood, many of the main results still seem unclear and at times contradictory. In these instances there is little discussion of why these seeming contradictions would be expected or biologically relevant either in the main results or in the discussion. And, follow up experiments designed to dissect apart these nuances are lacking. A few examples of these are listed below:

- The abstract states "we show that the asymmetric organization of the cytoskeleton is transmitted to the nuclear envelope and subsequently to nucleoplasmic proteins". However, the authors report that perturbing the cytoskeletal organization with blebbistatin, nocodazole, and cytochalasin does not alter nuclear polarization. This seems to directly contradict this earlier statement.
- The authors looked at multiple LEM domain-containing proteins including EMD, MAN1, and LAP2a. These proteins all polarized, though not in the same way and EMD knock-down affected only LAP2a. However, there is no discussion as to why EMD wouldn't affect MAN1 and if this is significant. How does this contribute to the proposed mechanism, and does this suggest a hierarchy of LEM domain-containing proteins in nuclear organization?
- EMD knock-down affected unphosphorylated Lamin A/C, but not phosphorylated Lamin A/C or

Lamin B. However there is no discussion or further experiments to determine why this may be occurring.

- Many members of the same family show different localization trends and responses to perturbations. E.G. SUN1 and SUN2 and nesprin-1 and nesprin-2. These proteins also showed different responses to EMD knockdown. These results are described, and the authors note that it's "interesting" that they show different trends. However, the reader is left wondering why and whether this is significant.

- In my original comments I noted: "In lines 107-114, the authors seem to suggest that sites of active transcription become more localized to the rear tip of the nucleus upon EMD knockdown, noting trends in nuclear actin and Ser5 phosphorylated RNAPII follow this trend. However, these observations appear to be in conflict with a concurrent trend of H3K27me3, an epigenetic mark of heterochromatin, as well localizing to the rear tip of the nucleus upon EMD knockdown". Although the language of this section has been toned down, the conflict inherent in these observations remains un-addressed. Active polymerase appears to re-localize to the rear of the nucleus. Facultative heterochromatin as well becomes more localized to the rear of the nucleus. Furthermore, a marker for active chromatin, H3K9ac, appears to become localized to the front of the nucleus upon treatment. While, I agree that this is largely outside of the main scope of this paper, the authors do choose to highlight the redistribution of nucleoplasmic proteins in the abstract, introduction and again in the discussion. As the authors, choose to highlight these results, I believe this paper would benefit from an interpretation of what appear, to me, to be conflicting results.

- While I appreciate that the authors tried to ensure that their results are not due to forcing cells to migrate on patterned lines, I am concerned that some of the results that are reported seem to vary whether the cells were randomly polarized or whether they were pre-patterned. One instance where this was of concern was in comparing the chromosome territories in patterned and non-patterned RPE-1 cells. In randomly polarized cells, the chromosomes showing significant territories were 12 and 18, however these were not significant in patterned cells, instead 3 was significant. If EMD is mediating this then the territories should be the same whether the polarity is stimulated or native, I would think. The other instance is in Supplementary Figure 6c where the size of focal adhesions were reduced in non-patterned RPE cells in EMD knock-down the difference is small compared to the effect in patterned RPE-1 cells.

- In my original comments, I noted: "The data pertaining to the chromosome territories do not appear to support the conclusions...". I thank the authors for heeding these suggestions and toning down the language. This said, I do not believe the authors have yet to provide sufficient evidence for A: asymmetric organization of chromatin in polarized cells. B: a role for EMD in a hypothetical asymmetric organization of chromatin. The only supporting data for either of these statements is the loss of localized enrichment in one out of the five examined chromosome territories and a DamID experiment, the results of which do not seem particularly informative.

- In my original comments, I noted: "In lines 112-113, the authors cite localization of H3K9ac and H3K4me3 in support of similarly distributed Ser5 phosphorylated RNAPII. However, the authors do not appear to include the data regarding localization of these histone marks in EMD knockdown cells. These experiments should be included.". I thank the authors for the additional time and effort to generate and include these distribution maps. I would recommend adding these additional distribution maps to the main figure rather than the supplementary. This would be especially helpful as the supplementary figure in which these additional distribution maps are included displays representative images rather than comparable distribution maps. That said, when comparing distribution maps from supplementary figure 2K and the distribution maps in figure 4H, RNAP localization does not appear distributed in a manner that is "...similar to the active chromatin markers H3K9ac and H3K4me3".

- In my original comments I asked: "Do other perturbations of cytoskeletal tractions (e.g. blebbistatin, actin targeting drugs, or perturbations to Rho GTPase) likewise perturb the organization of nuclear proteins like EMD and others?". I thank the authors for performing these additional experiments. The authors find that, although addition of these drugs do not completely abolish the EMD frontal distribution, three out of the four drugs do appear to significantly alter these distributions. The authors as well, find no concurrent significant alterations to the ER upon drug treatment. This is an interesting observation. Do the authors propose that that altering the force driven enrichment of EMD to the outer membrane is altering EMD's ability to enter the ER?

Minor comments and concerns:

- In supplementary movie 1, it is difficult to tell how the polarization of the cell is actually changing as the scale bar in the graph changes at every frame.
- I don't understand the first sentence in the second Results paragraph, "Emerin (EMD), an integral membrane protein of the NE, is one of the few proteins never reported to be present at both... from outside the nucleus to the inside."

Reviewer #3 (Remarks to the Author):

The authors have adequately addressed my concerns.

Point-by-point answer to reviewer's comments to manuscript "Front-rear polarity of the cell nucleus" (NCOMMS-19-22054A). :

Reviewer #1 (Remarks to the Author):

The authors addressed my criticisms comprehensively and added two additional cell manipulations to their assessment of nuclear polarity. I recommend publication in Nature Comm.

We thank the reviewer for this positive recommendation and for the work on our manuscript.

Reviewer #2 (Remarks to the Author):

In the revised manuscript "Front-rear polarity of the cell nucleus", the authors have added additional discussion and experimental data to address the my, and the other reviewer concerns. Although I believe that the results are interesting in their own right, **I still have substantial concerns regarding the formatting and conclusions of the manuscript.** There is an impressive amount of work presented here; however, the presentation of the data and the biological conclusions are lacking interpretation and support. My comments on the previous submission indicated this (e.g. a complete lack of introduction section) and the experimental suggestions were intended primarily to try and elucidate some of the mechanism behind the results presented (e.g. is this mediated by forces, cytoskeleton, or simply organelle location?). Testing this would require manipulating both forces, cytoskeletal, and organelle locations experimentally. Unfortunately, it appears that the results from these experiments were largely inconclusive. Or at least, are presented in such a way, as that I'm unable to determine the underlying biology driving the nuclear polarization. Further, the mathematical model presented is not experimentally tested in any substantive way.

We thank the reviewer for his/her work on our manuscript. We have revised the manuscript taking into account all the points raised, particularly taking care in toning down previous claims and providing detailed discussion on potential caveats in the interpretation of current data and open issues in need for follow up studies. We hope that the originality of our results, as appreciated by the other referees, will now stand out more clearly. We indeed produced an extensive and in-depth characterization of nuclear polarity. We apologize for the lack of clarity and we discussed better our new results in this revised version of the manuscript. Essentially, drug treatments are sufficient to perturb the cytoskeleton but not to disrupt cell polarity. Accordingly to our working model, to a biased distribution of the ER, it corresponds a biased distribution of elements at the NE, one of which is EMD. It is important to recall that the mathematical model we proposed has a descriptive intent and its aim is to show how simple assumptions could lead to the non-trivial phenotype that we observed.

Much of the manuscript is devoted to describing the experiments and results, but it is often left to the reader make sense of the significance and guess about the biological mechanism. Seemingly contradictory results are glossed over as being "interesting", but not explored further. **As a result, the major conclusions that I can draw from the manuscript are that in polarized cells, some nuclear proteins also become polarized and that EMD knock down affects some, but not all of these proteins and sometimes does so in opposing ways. The proteins that polarize and the susceptibility of these proteins to EMD knockdown are different in different cells.** I feel like there is the potential for an interesting story to be made from the work presented here, but as it

stands, **the experiments performed, and the description of the results, is not sufficient to warrant publication in Nature Communications.**

We are glad that the reviewer perfectly got the core of our message, although unfortunately he did not share our enthusiasm. Our first and essential conclusion is that in front-rear polarized cells some nuclear proteins are polarized too. This is probably the most important observation we did. Our second conclusion is that EMD is partially responsible for this polarization. Our data also clearly show that: 1) not all nuclear proteins are biased in polarized cell and 2) not all respond to EMD loss in the same manner or extent. We agree with the reviewer that much more has to be done to explore the relationship between cellular and nuclear polarity and explain the observations that now seem contradictory. This relationship already appears complex and is not unlikely that specific and sometimes redundant molecular mechanisms are responsible for the biased localization of different nuclear elements. Further studies are clearly needed to better elucidate this possible newly defined cellular function.

Major comments and concerns:

1) While I appreciate that the authors have reformatted the manuscript to be more easily understood, many of the main results still seem unclear and at times contradictory. In these instances there is little discussion of why these seeming contradictions would be expected or biologically relevant either in the main results or in the discussion. And, follow up experiments designed to dissect apart these nuances are lacking. A few examples of these are listed below:

- The abstract states “we show that the asymmetric organization of the cytoskeleton is transmitted to the nuclear envelope and subsequently to nucleoplasmic proteins”. However, the authors report that perturbing the cytoskeletal organization with blebbistatin, nocodazole, and cytochalasin does not alter nuclear polarization. This seems to directly contradict this earlier statement.

We thank the reviewer for having suggested the drug treatment experiments. We agree that the interpretation of the obtained results is not trivial. We improved the explanation of these new findings in the discussion of our revised version of our manuscript. All the drug treatments we tested clearly affected the cytoskeleton and the ER, as shown by the altered cell shape and KDEL distribution map. Nevertheless, as already pointed out, we found that ER and Golgi frontal localization was always preserved and, accordingly to our working model, this could be sufficient to preserve nuclear polarization. In agreement with the literature, we performed drug treatments using always sub-lethal concentrations for a short time. These conditions are sufficient to perturb the cytoskeleton but not to destroy the polarity of the cell, and indeed, we detected alterations of nuclear polarity, not its disappearance. Importantly, when we dramatically perturbed cell polarity, forcing RPE1 cells in a non-polarized state by plating them on a glass coated with poly-D-lysine and PLL-g-PEG (Fig.1 a&b), we finally abolished nuclear polarity (Fig.1 c). Please refer also to the original reply to the Reviewer#3.

- The authors looked at multiple LEM domain-containing proteins including EMD, MAN1, and LAP2 α . These proteins all polarized, though not in the same way and EMD knock-down affected only LAP2 α . However, there is no discussion as to why EMD wouldn't affect MAN1 and if this is significant. How does this contribute to the proposed mechanism, and does this suggest a hierarchy of LEM domain-containing proteins in nuclear organization?

We thank the reviewer for this comment. We agree that our data suggest a possible hierarchy of LEM domain-containing proteins in nuclear organization and now we addressed this more clearly in the main text. LEM domain-containing proteins have different and non-overlapping functions. EMD and Man1 are similar, they both carry a transmembrane domain and they localize at the nuclear envelope. Accordingly, in polarized cells, EMD and Man1 are both enriched toward the front of the cell. At the contrary, LAP2 α has no transmembrane domain and is indeed localized in the nucleoplasm. Moreover, in polarized cells, LAP2 α localization is complementary to EMD and is

Fig. 1: Induction of a non-polarized state in RPE-1 cells. *a* Representative cells on poly-D-lysine/PLL-g-PEG-coated glass (15 $\mu\text{g}/\text{ml}$ and 0.5 mg/ml, mixed 1:1), and fibronectin-coated glass (25 $\mu\text{g}/\text{ml}$). *b* Quantification of the speed of cells on poly-D-lysine/PLL-g-PEG-coated glass (15 $\mu\text{g}/\text{ml}$ and 0.5 mg/ml, mixed 1:1), and fibronectin-coated glass (25 $\mu\text{g}/\text{ml}$) $n_{\text{fibronectin}}=565$ cells, $n_{\text{PLL-g-PEG/Poly-D-lysine}}=449$ cells. *c* single cell (left panel) and distribution maps for the nucleus, Golgi and emerin in cells plated on poly-D-lysine/PLL-g-PEG-coated glass (15 $\mu\text{g}/\text{ml}$ and 0.5 mg/ml, mixed 1:1).

affected by EMD knock-down, while Man1 localization is not. Neither LAP2 α or Man1 knock-down perturbs EMD preferential localization. These data, as highlighted by the reviewer, suggest a hierarchy of LEM domain-containing proteins in nuclear polarization. All LEM domain-containing proteins share many interactors and the relationships between them could be complex. We could speculate that Man1 and EMD, that have similar behavior, reach the NE envelope through the same route. This would explain why one is not affected by the loss of the other. We could as well speculate that LAP2 α and EMD compete for binding partners at the nuclear envelope towards the front of the cell.

- EMD knock-down affected unphosphorylated Lamin A/C, but not phosphorylated Lamin A/C or Lamin B. However there is no discussion or further experiments to determine why this may be occurring.

We can speculate that when Lamin A/C is phosphorylated, it is not anymore found at the nuclear envelope, and could not interact with EMD or with other nuclear envelope proteins, hence losing its preferential localization. We now added this comment to the manuscript. It could be, that upon phosphorylation, lamin A/C loses its interaction with EMD and in consequence, is not enriched at the frontal area of NE anymore.

- Many members of the same family show different localization trends and responses to perturbations. E.G. SUN1 and SUN2 and nesprin-1 and nesprin-2. These proteins also showed

different responses to EMD knockdown. These results are described, and the authors note that it's "interesting" that they show different trends. However, the reader is left wondering why and whether this is significant.

We agree with the reviewer that this is an important point to clarify. Our study provides a characterization of nuclear polarity. Essentially, it measures the extent of polarity transmission from the cell to the nucleus. We made an effort (as appreciated by all reviewers) to perform this analysis as exhaustively as possible. The results depict a complex scenario in which members of the same family show different preferential localization as a function of the cell polarity axis. However, this complexity does not affect our main claim, which is that nuclear proteins localization depends (at least in some cases) on the polarity of the cell. Additionally, we proved that in some cases polarity is affected by EMD loss. The detailed investigation of specific molecular mechanisms driving each protein localization or re-localization upon EMD knock-down is very interesting, but beyond the scope of this work. Importantly, it is not our claim that EMD is the only protein responsible for nuclear polarity transmission. All our data point to the conclusion that clearly EMD plays an important role in polarity transmission from the cytoplasm to the nucleus, but it is clearly not the only player.

- In my original comments I noted: "In lines 107-114, the authors seem to suggest that sites of active transcription become more localized to the rear tip of the nucleus upon EMD knockdown, noting trends in nuclear actin and Ser5 phosphorylated RNAPII follow this trend. However, these observations appear to be in conflict with a concurrent trend of H3K27me3, an epigenetic mark of heterochromatin, as well localizing to the rear tip of the nucleus upon EMD knockdown". Although the language of this section has been toned down, the conflict inherent in these observations remains un-addressed. Active polymerase appears to re-localize to the rear of the nucleus. Facultative heterochromatin as well becomes more localized to the rear of the nucleus. Furthermore, a marker for active chromatin, H3K9ac, appears to become localized to the front of the nucleus upon treatment. While, I agree that this is largely outside of the main scope of this paper, the authors do choose to highlight the redistribution of nucleoplasmic proteins in the abstract, introduction and again in the discussion. As the authors, choose to highlight these results, I believe this paper would benefit from an interpretation of what appear, to me, to be conflicting results. We would like to recall that the spatial resolution of our maps, which should be considered as probability distributions, is not sufficient to infer or exclude the co-localization of different chromatin domains. We apologize to the reviewer that we did not explicitly discuss this possible caveat of our method in our original manuscript. We now provide this discussion in this revised version. The observation that on average both phosphorylated RNAPII (marker of active transcription) and H3K27me3 (an epigenetic mark of facultative heterochromatin) are enriched at the rear of the nucleus upon EMD knockdown (with respect to Ctrl) cannot be interpreted as a co-localization of the two. The rear (or the front) of the nucleus should be considered as a huge "macro-domain", where both regions of heterochromatin and euchromatin could be enclosed. Following the reviewer's advice, we further toned-down these claims.

- While I appreciate that the authors tried to ensure that their results are not due to forcing cells to migrate on patterned lines, I am concerned that some of the results that are reported seem to vary whether the cells were randomly polarized or whether they were pre-patterned. One instance where this was of concern was in comparing the chromosome territories in patterned and non-patterned RPE-1 cells. In randomly polarized cells, the chromosomes showing significant territories were 12 and 18, however these were not significant in patterned cells, instead 3 was significant. If EMD is mediating this then the territories should be the same whether the polarity is stimulated or native, I would think. The other instance is in Supplementary Figure 6c where the size of focal adhesions were reduced in non-patterned RPE cells in EMD knock-down the difference is small compared to the effect in patterned RPE-1 cells.

Cells self-polarize and migrate both on 2D or micro-patterned lines (1D). However, cellular and nuclear shapes change dramatically in these two conditions. The level of similarity between the results on 2D and 1D patterns were striking, and we do not found surprising that there are discrepancies between them. Moreover, is our claim that EMD is partially responsible for chromosome positioning but is not its only regulator. Cellular and nuclear shapes are too different between the experiments on 1D and 2D patterns to expect a perfect match on chromosomes positioning and the same dependency from EMD loss. Similarly, since on 1D patterns cells are more polarized, one could expect also that they are more contractile. This, we could speculate, is the reason of the different extent of the reduction of focal adhesion area upon EMD knock-down.

- In my original comments, I noted: “The data pertaining to the chromosome territories do not appear to support the conclusions...”. I thank the authors for heeding these suggestions and toning down the language. This said, I do not believe the authors have yet to provide sufficient evidence for A: asymmetric organization of chromatin in polarized cells. B: a role for EMD in a hypothetical asymmetric organization of chromatin. The only supporting data for either of these statements is the loss of localized enrichment in one out of the five examined chromosome territories and a DamID experiment, the results of which do not seem particularly informative.

We agree with the reviewer that in the first version of the manuscript we over-interpreted our results and accordingly we toned-down our conclusions. The current version of the text correctly reports that only one over the six tested chromosomes shows a biased distribution. We do not agree that the DamID experiment is not informative. Indeed, it shows that, together with EMD, also the chromatin region interacting with it is biased toward the front of the cell. This evidence supports that the cellular polarity can be transmitted to the chromatin.

- In my original comments, I noted: “In lines 112-113, the authors cite localization of H3K9ac and H3K4me3 in support of similarly distributed Ser5 phosphorylated RNAPII. However, the authors do not appear to include the data regarding localization of these histone marks in EMD knockdown cells. These experiments should be included.”. I thank the authors for the additional time and effort to generate and include these distribution maps. I would recommend adding these additional distribution maps to the main figure rather than the supplementary. This would be especially helpful as the supplementary figure in which these additional distribution maps are included displays representative images rather than comparable distribution maps. That said, when comparing distribution maps from supplementary figure 2K and the distribution maps in figure 4H, RNAP localization does not appear distributed in a manner that is “...similar to the active chromatin markers H3K9ac and H3K4me3”.

We thank the reviewer for this suggestion and we moved those maps to the main Fig. 2h and Supplementary Fig. 2f. Additionally, we improved their description and explanation in the text.

- In my original comments I asked: “Do other perturbations of cytoskeletal tractions (e.g. blebbistatin, actin targeting drugs, or perturbations to Rho GTPase) likewise perturb the organization of nuclear proteins like EMD and others?”. I thank the authors for performing these additional experiments. The authors find that, although addition of these drugs do not completely abolish the EMD frontal distribution, three out of the four drugs do appear to significantly alter these distributions. The authors as well, find no concurrent significant alterations to the ER upon drug treatment. This is an interesting observation. Do the authors propose that that altering the force driven enrichment of EMD to the outer membrane is altering EMD’s ability to enter the ER?

We thank the reviewer for this comment and we clarified it in the revised manuscript. In figure 4f we show that drug treatments indeed affect the ER, as clearly visible by KDEL distribution maps. However, the perturbations induced by different drugs are sufficient to alter the ER distribution but not to impede its preferential localization toward the front of the cell. This is the reason why,

accordingly to our model, the drugs we tested were not sufficient to affect the preferential localization of EMD.

Minor comments and concerns:

- In supplementary movie 1, it is difficult to tell how the polarization of the cell is actually changing as the scale bar in the graph changes at every frame.

We fixed the plot scale in the Supplementary Movie 1.

- I don't understand the first sentence in the second Results paragraph, "Emerin (EMD), an integral membrane protein of the NE, is one of the few proteins never reported to be present at both... from outside the nucleus to the inside."

We apologize for the lack of clarity. We changed this sentence to: "Emerin (EMD), an integral membrane protein of the NE, has been reported to be present at both, the inner (INM) and the outer nuclear membrane (ONM)²¹, hence potentially enabling polarity transmission from the cytoplasm to the nucleoplasm."

Reviewer #3 (Remarks to the Author):

The authors have adequately addressed my concerns.

We are grateful to the reviewer for this positive outcome.

REVIEWERS' COMMENTS:

Reviewer #2 (Remarks to the Author):

I thank the authors for their point-by-point response to my previous comments. At this point, I'll defer to the other reviewers and editor that the current work is suitable for publication in Nat Comm. There are an impressive amount of experiments and data included here that describe how various nuclear proteins respond to cell polarity and Emerin knockdown. I look forward to follow up manuscripts that expand upon the biological function of this nuclear polarization and the mechanism by which this is mediated by EMD.